# Semantic Alignment for Prompt-Tuning in Vision Language Models

**Hari Chandana Kuchibhotla**[‡]                              *ai20resch11006@iith.ac.in*
*Indian Institute of Technology Hyderabad, India*
**Sai Srinivas Kancheti**[‡]                                 *cs21resch01004@iith.ac.in*
*Indian Institute of Technology Hyderabad, India*
**Abbavaram Gowtham Reddy**[*]                       *gowtham.abbavaram@cispa.de*
*CISPA Helmholtz Center for Information Security, Saarbrücken, Germany*
**Vineeth N Balasubramanian**                              *vineethnb@iith.ac.in*
*Indian Institute of Technology Hyderabad, India*

**Reviewed on OpenReview:** *https: // openreview. net/ forum? id= avDr56QjSI*

## Abstract

Going beyond mere fine-tuning of vision-language models (VLMs), learnable prompt tuning has emerged as a promising, resource-efficient alternative. Despite their potential, effectively learning prompts faces the following challenges: (i) training in a low-shot scenario results in overfitting, limiting adaptability, and yielding weaker performance on newer classes or datasets; (ii) prompt-tuning's efficacy heavily relies on the label space, with decreased performance in large class spaces, signaling potential gaps in bridging image and class concepts. In this work, we investigate whether better text semantics can help address these concerns. In particular, we introduce a prompt-tuning method that leverages class descriptions obtained from Large Language Models (LLMs). These class descriptions are used to bridge image and text modalities. Our approach constructs part-level description-guided image and text features, which are subsequently aligned to learn more generalizable prompts. Our comprehensive experiments conducted across 11 benchmark datasets show that our method outperforms established methods, demonstrating substantial improvements.

## 1 Introduction

Foundational Vision-Language Models (VLMs) like CLIP (Radford et al., 2021) and ALIGN (Jia et al., 2021) have displayed remarkable zero-shot and open-vocabulary capabilities in recent years. This has led to VLMs being employed in various vision-only downstream tasks such as open-vocabulary image classification (Liang et al., 2022), object detection (Feng et al., 2022), and image segmentation (Lüddecke & Ecker, 2022). Trained on extensive web data, these models often use a contrastive loss to align image-text pairs in a shared embedding space, allowing them to represent diverse concepts.

Recently, learnable prompt-tuning (Zhou et al., 2021; Khattak et al., 2023; Fahes et al., 2023) has emerged as a promising parameter-efficient alternative for fine-tuning foundation models. Prompt-tuning methods introduce additional learnable parameters called *prompt vectors*, which are tuned on task-specific data. This approach adapts VLMs for a specific downstream task without affecting the pre-trained parameters of the VLM. While prompt-tuning methods have shown great promise, efficiently learning prompt vectors faces the following challenges: (i) training prompts in a low-shot setting leads to overfitting, hindering their generalizability, and exhibiting sub-optimal performance when applied to newer classes or datasets (Shi & Yang, 2023; Khattak et al., 2023; 2022), (ii) the performance of prompt-tuning methods can be highly dependent on the label space used for classification. During inference, if the label space is large, the performance tends to

---

[‡]Equal contribution, [*] Work done at IIT-Hyderabad

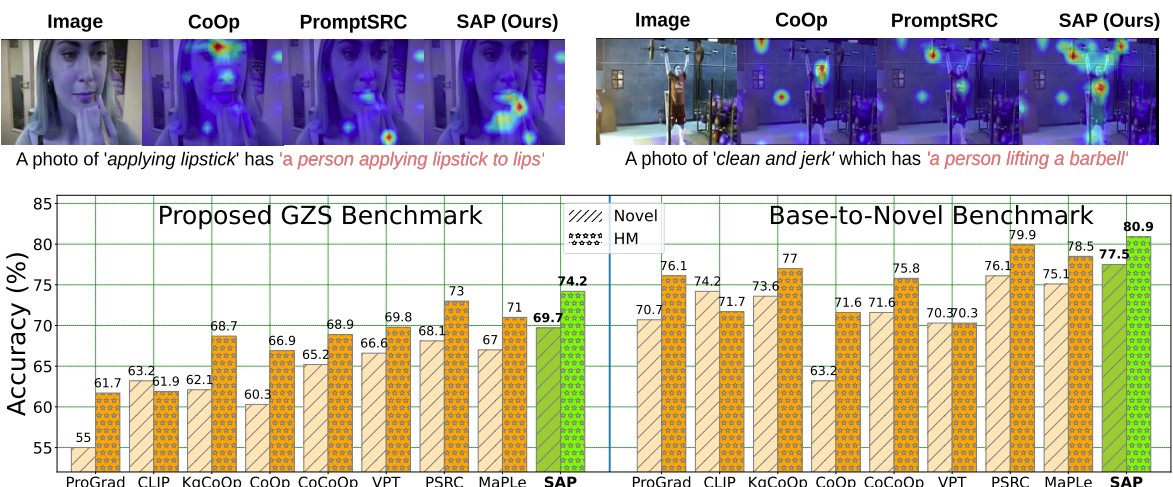

Figure 1: **Top:** Comparison of GradCAM (Selvaraju et al., 2017) visualizations for our proposed method SAP against other baselines, on classes *"Applying Lipstick"* and *"Clean and Jerk"* from an Action Recognition dataset (Soomro et al., 2012). The saliency maps indicate image regions that are most relevant to the descriptions "A photo of applying lipstick has a person applying lipstick to lips" and "A photo of clean and jerk which has a person lifting a barbell" respectively. SAP effectively localizes the text semantics in images compared to baselines. **Bottom:** SAP surpasses other baselines on Generalized Zero-Shot (GZS) and Base-to-Novel (B2N) benchmarks, showing improvements of **+1.6%** and **+1.2** on Novel Accuracy and Harmonic Mean (HM) for GZS, and **+1.4%** and **+0.9** for B2N compared to best performing baselines.

decrease due to bias towards the seen classes the model was fine-tuned on (see empirical evidence in Tab. 2 of § 5). These issues indicate that there is a lack of understanding of images and classes based on their detailed semantic components. For example, an image of a cat should be understood through its specific features like 'whiskers' and 'tail', not just the class name 'cat.' To address this, we propose *SAP* (**S**emantic **A**lignment for **P**rompt-tuning), which uses class descriptions to learn generalizable prompts. Our semantic alignment module brings together class descriptions for class names with the text modality and the image modality using a cross-attention mechanism and a learnable bias. This alignment helps the model grasp the relationship between different parts of an image and their textual descriptions, leading to a more detailed and accurate representation.

Our method uses class descriptions to guide the creation of such image and text features that correspond to specific parts or aspects of a class. However, we observe that merely using class descriptions alone does not address the challenges presented above, as shown in Tab. 6 of § 5.2. We demonstrate that careful *semantic alignment* between image and text features is crucial for effectively leveraging class descriptions. Given a set of class descriptions, we show how to construct *description-guided* image and text features. For instance, for an image of a cat, and a class description 'has a large tail', the corresponding description-guided image feature encodes the part-level visual semantic information related to the description, using a cross-attention mechanism and a learnable bias. The description-guided text features constitute the class name along with the semantics of each class (obtained from the descriptions), resulting in information-rich and robust text features. We then compute semantic alignment as the average cosine similarity between description-guided image and text features, for relevant descriptions. Note that our approach includes the alignment between both improved text and improved image features, making our approach multimodal. We use pre-trained Large Language Models (LLMs) to generate class descriptions in an inexpensive manner. A recent set of works (Menon & Vondrick, 2023; Yang et al., 2022) has shown that class descriptions obtained from LLMs can be naively used to classify images on a given dataset with fixed categories. We go beyond and leverage these class descriptions to perform low-resource prompt-tuning, and show that such adapted VLMs show better generalization to unseen, novel classes. Fig. 1 illustrates the effectiveness of SAP over other baselines on two benchmarks, Generalized Zero-Shot Classification (GZS) and Base-to-Novel Classification (B2N), defined in § 5. As our semantic alignment is part-level, SAP showcases superior localization of visual concepts relevant to a class description, as seen through class activation maps, when compared to other baselines.

| Method | Text Prompts | Image Prompts | Use of External Knowledge | Part-level img-text alignment | Evaluation Benchmarks | # of Additonal Trainable Parameters |
|---|---|---|---|---|---|---|
| **CoOp** [IJCV '22], **CoCoOp** [CVPR '22], **KgCoOp** [CVPR '23] **ProGrad** [ICCV '23], **ProDA** [CVPR '22] | ✓ | ✗ | ✗ | ✗ | B2N, XDataset, DG | 2k - 36k |
| **MaPLe** [CVPR '23], **PSRC** [ICCV '23], **LoGoPrompt** [ICCV '23] | ✓ | ✓ | ✗ | ✗ | B2N, XDataset, DG | 36k - 3.55M |
| **KAPT** [ICCV '23], **CoPrompt** [ICLR '24], **CLIP-VDT** [ICCVW '23] | ✓ | ✓ | ✓ | ✗ | B2N, XDataset, DG | 1.3M - 4.74M |
| **SAP (Ours)** | ✓ | ✓ | ✓ | ✓ | GZS, B2N, XDataset, DG, Classification without Class-names | **36K** |

Table 1: Comparison of the proposed method, SAP, with other related work on various key aspects involving fine-tuning VLMs for better generalization. B2N: Base-to-Novel, XDataset: Cross Dataset, DG: Domain Generalization, GZS: Generalized Zero-Shot.

Tab. 1 delineates the key differentiators of our approach compared to other baselines. Most existing prompt-tuning methods do not use additional text semantics; even among the recent few that use such information, our method utilizes class descriptions at part-level for both image and text. This strategy leads to non-trivial performance enhancements across benchmark datasets and improved localizations in novel classes or datasets. Additionally, we highlight a gap in the evaluation scheme used in existing prompt-tuning efforts, which demonstrate the performance of learned prompts primarily on the tasks of Base-to-Novel classification and cross-dataset evaluation. Inspired by the traditional Generalized Zero-Shot Learning (G-ZSL) (Xian et al., 2017; Liu et al., 2023) paradigm, we posit that generalization in the zero-shot setting is more realistic when considering both base and novel classes at inference. We call this protocol *GZS evaluation* – the first such effort among prompt-tuning methods. We also propose another benchmark – *Classification without Class-names* – where the method is exclusively evaluated using class descriptions to classify images when its label lies outside CLIP's vocabulary. Our contributions are:

- We propose a prompt-tuning method to fine-tune VLMs that can leverage class descriptions obtained from an LLM. Our semantic alignment module allows for integration of class descriptions obtained from class names with both text modality and image modalities using a cross-attention mechanism and a learnable bias, thus bridging the two modalities. This improved alignment allows us to learn prompts that can generalize well to unseen classes and datasets.

- We carry out a comprehensive suite of experiments with comparisons against state-of-the-art and very recent methods on eleven standard benchmark datasets. We outperform existing baselines with a significant margin on all evaluation protocols.

- We propose two new evaluation protocols: GZS evaluation and Classification without Class-names to better study the generalizability of prompt-tuning methods for VLMs. Our method consistently outperforms earlier baselines on these protocols, too.

## 2 Related Work

**Vision-Language Models.** Vision-language models (VLM) exhibit significant promise in acquiring generic visual representations. VLMs aim to harness natural language guidance for image representation learning and concurrently align both the text and image features within a shared embedding space. We consider encoder-only VLMs which comprise of three components: a text encoder, an image encoder, and a learning methodology that effectively utilizes information from both text and image modalities. Recent research on learning transferable visual representations delves into establishing semantic connections between text and visual elements, capitalizing on a vast reservoir of internet-based image-text pairs. For instance, CLIP (Radford et al., 2021) is the product of contrastive learning from 400 million image-text pairs, while ALIGN (Jia et al., 2021) utilizes 1.8 billion noisy image-text pairs extracted from raw alt-text data. Nonetheless, a substantial challenge persists in transferring these foundational models to downstream tasks while preserving their initial generalization capabilities. To address this, we use auxiliary information in the form of class descriptions to better align image and text features, enhancing the model's performance and generalizability.

**Prompt-Tuning.** Prompt-tuning introduces task-specific text tokens designed to be learnable to customize the pre-trained VLM for downstream tasks. Context Optimization (CoOp) (Zhou et al., 2021) marks the pioneering effort in replacing manually crafted prompts with adaptable soft prompts, fine-tuned on labeled

few-shot samples. Conditional Context Optimization (CoCoOp) (Zhou et al., 2022) builds upon this by generating image-specific contexts for each image and merging them with text-specific contexts for prompt-tuning. In contrast, Visual Prompt Tuning (Jia et al., 2022) introduces learnable prompts exclusively at the vision branch, resulting in sub-optimal performance for transferable downstream tasks. ProDA (Lu et al., 2022) focuses on learning the distribution of diverse prompts. KgCoOp (Yao et al., 2023) introduces regularization to reduce the discrepancy between learnable and handcrafted prompts, enhancing the generalizability of learned prompts to unseen classes. PSRC (Khattak et al., 2023) shares a similar concept with KgCoOp (Yao et al., 2023) but introduces Gaussian prompt aggregation. ProGrad (Zhu et al., 2023) selectively modifies prompts based on gradient alignment with a hard-coded prompt. MaPLe (Khattak et al., 2022) introduces prompts at text and image encoder branches and link them with a coupling function. In a different approach, LoGoPrompt (Shi & Yang, 2023) capitalizes on synthetic text images as effective visual prompts, reformulating the classification problem into a min-max formulation. Although these methods have shown promising results, they suffer from overfitting to the training classes when trained in a low-shot manner. This overfitting limits their generalizability and results in sub-optimal performance on newer classes or datasets. We address this issue by leveraging external information in the form of class descriptions to semantically align image and text features, helping us learn generalizable prompts.

**Use of External Knowledge.** A set of recent works (Menon & Vondrick, 2023; Yang et al., 2022; Pratt et al., 2022) provide evidence that visual recognition can be improved using concepts, and not just class names. However, (Menon & Vondrick, 2023; Pratt et al., 2022) does not facilitate a way to perform fine-tuning on a downstream dataset. In contrast, (Yang et al., 2022) is a concept bottleneck model with a fixed label space and thus cannot be used for zero-shot classification. In fine-tuning methods incorporating external knowledge, KAPT (Kan et al., 2023) introduces complementary prompts to simultaneously capture category and context but lacks semantic alignment of each class description at the part-level of both image and text. On the other hand, CLIP-VDT (Maniparambil et al., 2023) utilizes semantic-rich class descriptions only in the text modality, without semantic alignment with images. In CoPrompt(Roy & Etemad, 2024), class descriptions are utilized via a regularizer acting as a consistency constraint to train the text prompts. There is no consideration of explicit semantic alignment with the image modality. In contrast, our approach utilizes class descriptions to semantically construct both text and image features, enhancing part-level alignment between the two modalities. This improved alignment helps us learn prompts that can generalize well to unseen classes and datasets. A comparison of our method with existing works is shown in Tab. 1.

## 3 Preliminaries and Background

VLMs perform image classification on a downstream dataset by comparing an image representation with text representations of the class names in the dataset's label space. When a small amount of labeled data is available, it has been shown that fine-tuning VLMs substantially boosts downstream performance (Zhou et al., 2021; 2022). However, the fine-tuned model does not generalize to novel classes that were absent during fine-tuning (Zhou et al., 2022). In this work, we propose **S**emantic **A**lignment for **P**rompt Learning (SAP), that leverages class descriptions to fine-tune VLMs for better generalization to novel classes. Before we describe our methodology, we briefly discuss the required preliminaries, beginning with CLIP (Radford et al., 2021), the VLM chosen as our backbone following earlier work (Zhou et al., 2021; 2022; Khattak et al., 2022; Lu et al., 2022; Yao et al., 2023; Khattak et al., 2023; Zhu et al., 2023). A summary of notations and terminology is presented in Appendix § A.

**CLIP Preliminaries.** CLIP consists of an image encoder $\theta$ and a text encoder $\phi$, which are trained contrastively on paired image-text data to learn a common multi-modal representation space. $\theta$ takes an image $\mathbf{x}$ as input and returns the image feature $\theta(\mathbf{x}) \in \mathbb{R}^d$. $\phi$ processes a text string $S$ into a $d$-dimensional feature vector $\phi(S) \in \mathbb{R}^d$. CLIP is trained with InfoNCE loss (van den Oord et al., 2018) to enhance cosine similarity for matching image-text pairs and to reduce it for non-matching pairs.

CLIP performs zero-shot recognition of an image $\mathbf{x}$ by choosing the most similar class name from a set of candidate class names $\mathcal{Y}$, i.e., predicted class $\hat{y} = \arg\max_{y \in \mathcal{Y}} sim(\theta(\mathbf{x}), \phi(y))$, where the similarity $sim$ is cosine-similarity. In practice, for a class name $y$, $\phi(y)$ is the text representation of a manually crafted prompt

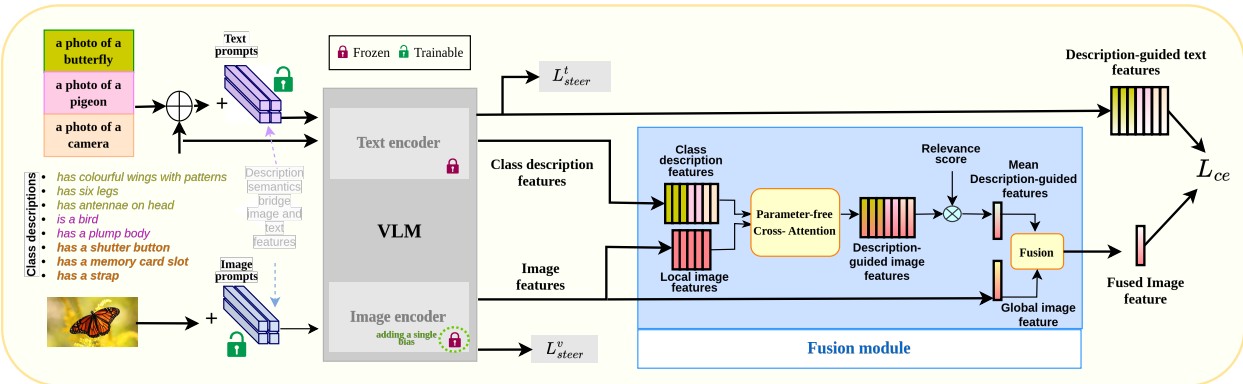

Figure 2: Our proposed workflow, SAP, performs part-based semantic alignment between image and text features. SAP integrates class descriptions into the text template which are passed through the text encoder to construct description-guided text features. Global and local image features are obtained from the image encoder. Description-guided image features are obtained by performing parameter free cross-attention between class descriptions and local features. These image features are pooled into a mean description-guided image feature, which is then fused with the global image feature to obtain the fused image feature. Description-guided text features and the fused image feature contain part-level semantic information, and are semantically aligned. We optimize a cross-entropy loss $L_{ce}$, and two steering losses $L_{steer}^{v}$, and $L_{steer}^{t}$.

encapsulating $y$ such as 'a photo of a [y]'. Zero-shot classification performance significantly depends on the label set $\mathcal{Y}$ considered, and varies with the template of the text prompt (Radford et al., 2021).

**Fine-Tuning CLIP with Learnable Prompts.** To perform efficient adaptation under limited supervision, prompt-tuning methods add a small number of learnable tokens to the input token sequence of either modality which are fine-tuned to generate task-specific representations. For instance, CoOp (Zhou et al., 2021) adds $n$ learnable text-prompts $\rho_t = \{\mathbf{p}_1^t, \ldots, \mathbf{p}_n^t\}$ to the token embeddings $\{\mathbf{w}_1^S, \ldots, \mathbf{w}_q^S\}$ of some text $S$. The final sequence $\{\mathbf{p}_1^t, .., \mathbf{p}_n^t, \mathbf{w}_1^S, .., \mathbf{w}_q^S\}$ is passed through $\phi$ to obtain the *prompted text feature* $\phi_p(S)^{\ddagger}$. We follow IVLP (Rasheed et al., 2022), which adds learnable prompt tokens at transformer layers of both image and text encoders. That is, along with text prompts, IVLP appends learnable visual prompts $\rho_v$ to patch tokens of image $\mathbf{x}$, which are passed through $\theta$ to yield the *prompted visual feature* $\theta_p(\mathbf{x})$. Let $\rho = \{\rho_t, \rho_v\}$ denote the set of all trainable text and visual prompts. These prompts are trained to maximize the similarity between a prompted image feature and the corresponding prompted text feature of its class label. Given $B$ image-text pairs $\{(\mathbf{x}_i, y_i)\}_{i=1}^{B}$, where $y_i \in \mathcal{Y}$, the likelihood of $\mathbf{x}_i$ predicting class $y_i$ is given by $\mathbb{P}_\rho(y_i \mid \mathbf{x}_i) = \dfrac{\exp(sim(\theta_p(\mathbf{x_i}), \phi_p(y_i))/\tau)}{\sum\limits_{y \in \mathcal{Y}} \exp(sim(\theta_p(\mathbf{x_i}), \phi_p(y))/\tau)}$, where $\tau$ is the temperature and *sim* is cosine similarity. The negative log-likelihood loss to be optimized is $L(\rho) = \frac{-1}{B} \sum\limits_{i \in [B]} \log(\mathbb{P}_\rho(y_i \mid \mathbf{x}_i))$.

With the above background, we now present our methodology to use class descriptions to learn prompts that helps VLMs generalize better to unseen, novel classes.

## 4 Semantic Alignment for Prompt-tuning: Methodology

Given labeled data, most existing methods learn prompts that largely limit themselves to incorporating text information in the form of class labels only. We propose SAP, **S**emantic **A**lignment for **P**rompt-tuning, which utilizes auxiliary information in the form of class descriptions obtained from LLMs to learn more generalizable prompts. Our method constructs description-guided image and text features that are semantically aligned with each other. Specifically, a class description provides a semantic context, and the corresponding description-guided image or text feature encodes part-level information related to this

---

‡We add a subscript $p$ to indicate prompted features for images and text

description. Our semantic alignment module allows for integration of class descriptions obtained from class names with both text modality and image modalities using a cross-attention mechanism and a learnable bias. This external semantic knowledge, derived from class descriptions, transfers to novel classes because the semantics represent common concepts shared across multiple classes, such as 'large tail' or 'whiskers'. An overview of our methodology is shown in Figure 2. We begin by describing how class descriptions are generated using LLMs.

## 4.1 Generating Class Descriptions

Large language models (LLMs) act as vast knowledge corpora that can be queried for the semantics of real-world objects. We use the popular LLM GPT-3.5 (Hagendorff et al., 2022) to obtain text descriptions for each class in a given dataset. Class descriptions commonly contain visual cues such as shape, texture, and color, as well as narratives of objects commonly correlated with the class. To keep our method cost-efficient, we use descriptions that are class-specific but not image-specific, thus making them reusable for a set of image samples (note that this is done only once per class label). We use the responses from the LLM as they are, and do not manually curate or filter them any further. This keeps our approach low-cost while integrating finer semantic details into fine-tuning of VLMs. Some examples of our class descriptions are provided in Appendix § E.

**Class Description Features.** For each class $y \in \mathcal{Y}$, where $\mathcal{Y}$ is the label space under consideration, we denote by $A_y$ the set of generated class descriptions. Let $A = \bigcup_{y \in \mathcal{Y}} A_y$, $N = |A|$ denote the set of descriptions of all classes and the size of the set, respectively. *Class description features* $\phi(A) \in \mathbb{R}^{N \times d}$ are obtained by passing the class descriptions through text-encoder $\phi$. In the following sections, we describe how SAP leverages class descriptions to construct description-guided image and text features, enabling us to learn prompts that generalize well.

## 4.2 Leveraging Class Descriptions for Text Features

The text feature $\phi(y)$ for a class $y \in \mathcal{Y}$ is generally obtained by encapsulating the class name in a text template, for eg. 'a photo of a [y]', and passing it through $\phi$. When class descriptions $A_y$ are given, we append them to the text template to generate $|A_y|$ distinct templates. For example for class $y = cat$ and $A_y = \{$'has whiskers', 'has a large tail'$\}$, we generate 2 description-guided templates 'a photo of a cat which has whiskers' and 'a photo of a cat which has a large tail'.

The description-guided templates are passed through text-encoder $\phi$ to generate description-guided text features $\phi(y; A_y) \in \mathbb{R}^{|A_y| \times d}$ for class $y$. For an image $\mathbf{x}$, the semantic alignment $\xi$ between the image feature $\theta(\mathbf{x})$ and description-guided text features for class $y$ is given by:

$$\xi(\theta(\mathbf{x}), \phi(y; A_y)) = \frac{1}{|A_y|} \sum_{a \in A_y} sim(\theta(\mathbf{x}), \phi(y; a)) \tag{1}$$

This simple way of incorporating class descriptions into the text modality works well in practice. We validate our design choices in Tab. 6 by comparing against alternative ways to incorporate class descriptions.

## 4.3 Leveraging Class Descriptions for Image Features

As shown in the fusion module of Fig. 2, we also leverage the class descriptions in the visual modality by first generating *description-guided* image features, and then fusing them with the *global* image feature. We describe this process below.

**Constructing Description-Guided Image Features.**

An image $\mathbf{x}$ is passed through the image encoder $\theta$ (which is a vision transformer in this section) and the output of the final transformer block of shape $(1 + 196 + n) \times d'$ is collected. Here, 1 corresponds to the $\mathbf{cls}_{\mathcal{I}}$ token, $n$ is the number of learnable prompt tokens, and $d'$ is the dimension of the transformer layer. In all earlier works, including CLIP, the $\mathbf{cls}_{\mathcal{I}}$ output token is passed through the final projection layer $proj \in \mathbb{R}^{d' \times d}$ of $\theta$ to

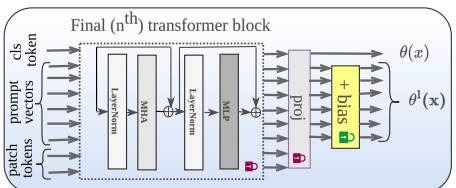

Figure 3: Addition of a bias vector to the last transformer block in $\theta$

obtain the *global image feature* $\theta(\mathbf{x}) \in \mathbb{R}^d$. These features capture the global context of the image but may not capture local object-level semantics (Rao et al., 2021). We aim to utilize the rich part-level local information hidden in the 196 patch tokens and establish their association with class descriptions. To obtain the *local image features* $\theta^l(\mathbf{x}) \in \mathbb{R}^{196 \times d}$, we pass the patch tokens through *proj* and add a learnable *d*-dimensional bias offset as shown in Fig. 3. This bias is added to fine-tune *proj* with local information, which otherwise is used only to obtain global image features from the last transformer block. The added learnable parameter, though small, effectively fine-tunes local features based on the descriptions, enhancing the quality of local image features.

We obtain description-guided image features by performing a parameter free cross-attention with class description features as queries, and local image features as both keys and values.

$$\theta^{desc}(\mathbf{x}) = CrossAttention(Q = \phi(A), K = \theta^l(\mathbf{x}), V = \theta^l(\mathbf{x})) \quad (2)$$

Here, $\phi(A) \in \mathbb{R}^{N \times d}$ are the class description features for all class descriptions $A$, $\theta^l(\mathbf{x}) \in \mathbb{R}^{196 \times d}$ are the local features of an image. The description-guided image features $\theta^{desc}(\mathbf{x}) \in \mathbb{R}^{N \times d}$ encode part-level local information relevant to the $N$ descriptions. For any description, the cross-attention module computes a weighted combination of the 196 local features, where the weights are determined by the similarity between the image patch and the description. Note that we obtain $N$ description features, one per description, for a single image. Since descriptions are common across classes and even datasets, these features contain information that can transfer to novel classes. Note that, similar to other works (Khattak et al., 2023; 2022), we only require a single forward pass to compute $\theta$ and $\theta_{desc}$. Usually, only the `cls` token is used for image features. We leverage the remaining tokens (instead of discarding them) to compute local image features $\theta_l$, and consequently $\theta_{desc}$.

**Fusing Description-Guided Features with Global Image Feature.** The description-guided image features described above use class descriptions from all classes, and not just the ground-truth class of the image. Since the class descriptions generated by LLMs may be noisy, not all descriptions are relevant to a specific image. To address this, we introduce a *relevance score* $\mathbf{r} \in [0,1]^N$, which quantifies each description's similarity to the image. This is computed as:

$$\mathbf{r} = softmax(\phi(A) \cdot \theta(\mathbf{x})) \quad (3)$$

We perform a weighted average of $\theta^{desc}(\mathbf{x})$ with $\mathbf{r}$, and obtain the *mean description-guided* feature $\bar{\theta}^{desc}(\mathbf{x}) \in \mathbb{R}^d$, which captures finer contexts in an image and is computed as:

$$\bar{\theta}^{desc}(\mathbf{x}) = \theta^{desc}(\mathbf{x})^\intercal \cdot \mathbf{r} \quad (4)$$

For an image, the global image feature $\theta(\mathbf{x}) \in \mathbb{R}^d$ encodes class information pertaining to the image and the mean description-guided feature $\bar{\theta}^{desc}(\mathbf{x}) \in \mathbb{R}^d$ encodes part-level visual context. We perform a fusion of both these features to yield the final *fused image feature* $\hat{\theta}(\mathbf{x})$.

$$\hat{\theta}(\mathbf{x}) = (1 - \alpha) \cdot \theta(\mathbf{x}) + \alpha \cdot \bar{\theta}^{desc}(\mathbf{x}) \quad (5)$$

We give a higher weight $\alpha \in [0,1]$ to the part-level features $\bar{\theta}^{desc}(\mathbf{x})$ of an image if the descriptions attend strongly to specific patches of the image. For each description, we consider the maximum attention weight over image patches as the specificity of the description. We then define $\alpha$ as the average specificity for all descriptions. We note that $\alpha$ is not a hyperparameter. This indicates the specificity of certain descriptions to some parts of the image. To see this, consider the case of a background image. Clearly such an image is uninformative w.r.t any class description, and its part-level features can be discounted. For each description, the maximum attention weight over image patches is a proxy for the specificity of the description. We then define $\alpha$ as the average specificity for all descriptions. The fused image feature $\hat{\theta}(\mathbf{x}) \in \mathbb{R}^d$ contains global visual semantics as well as part-level semantics.

## 4.4 Description-Guided Semantic Alignment

Given an image $\mathbf{x}$, we obtain the fused image feature $\hat{\theta}(\mathbf{x})$ as described in § 4.3. For every class $y \in \mathcal{Y}$, we obtain the description-guided text features $\phi(y; A_y)$ as described in § 4.2. We denote the learnable prompt

vectors by $\rho$, and we represent prompted features with subscript $p$. For instance, the prompted fused image feature is $\hat{\theta}_p(\mathbf{x})$, and so on. Prompts are trained by minimizing the negative log-likelihood of the training data $\{(\mathbf{x}_i, y_i)\}_{i=1}^B$:

$$L_{ce}(\rho) = -\frac{1}{B} \sum_{i=1}^{B} \log \frac{\exp(\xi(\hat{\theta}_p(\mathbf{x_i}), \phi_p(y_i; A_{y_i}))/\tau)}{\sum_{y \in \mathcal{Y}} \exp(\xi(\hat{\theta}_p(\mathbf{x_i}), \phi_p(y; A_y))/\tau)} \text{where } \xi(\hat{\theta}_p(\mathbf{x}), \phi_p(y; A_y)) = \frac{1}{|A_y|} \sum_{a \in A_y} sim(\hat{\theta}_p(\mathbf{x}), \phi_p(y; a))$$

(6)

where $\tau$ is the temperature parameter, and *sim* is cosine similarity. To compute semantic alignment $\xi$, we aggregate similarity between the fused image feature and the description-guided text feature over all pertinent class descriptions and normalize by their count. A relevant description in the image enhances its similarity to the class; however, the absence of a description in the image does not penalize its similarity to the class. Following (Yao et al., 2023; Khattak et al., 2023), we add regularization terms designed to penalize prompted features that deviate significantly from their unprompted counterparts. We use the *L1* penalty to regularize global image features and description guided text features.

$$L_{steer}^v(\rho) = \frac{1}{B} \sum_{i=1}^{B} \|\theta_p(\mathbf{x_i}) - \theta(\mathbf{x_i})\|_1 \qquad L_{steer}^t(\rho) = \frac{1}{|\mathcal{Y}|} \sum_{y \in \mathcal{Y}} \|\phi_p(y; A_y) - \phi(y; A_y)\|_1 \qquad (7)$$

The final objective is $\mathcal{L}(\rho) = L_{ce}(\rho) + \lambda_1 L_{steer}^v(\rho) + \lambda_2 L_{steer}^t(\rho)$, where $\lambda_1$ & $\lambda_2$ are hyperparameters.

**Inference:** Let $\mathcal{Y}'$ be the inference time label space, and $A_z$ be the class descriptions of class $z \in \mathcal{Y}'$. Using the learned prompt $p$, we compute the prompted fused image feature and the description-guided text features for all classes in $\mathcal{Y}'$. The class with the highest semantic alignment $\xi(\hat{\theta}_p(\mathbf{x}'), \phi_p(z; A_z))$ is then predicted as the final label. The overall algorithm of SAP is presented in Appendix § B .

## 5 Experiments and Results

We comprehensively evaluate the generalization performance of SAP on two newly proposed benchmarks – (i) Generalized Zero-Shot Classification (GZS) and (ii) Classification without Class-names (CwC) and existing benchmarks (iii) Base-to-Novel Generalization (B2N) and (iv) Cross-Dataset Generalization.

**Proposed Evaluation Benchmarks:**

**(i) Generalized Zero-Shot Classification (GZS).** In GZS, the label space of a dataset is equally split into disjoint base and novel classes. Only a small number (e.g., 16-shot) of labeled samples from the base classes are available as training data. However, during evaluation, the classification label space is the union of base and novel classes. As explained in § 3, zero-shot classification performance depends on the label space considered, and introducing the union of base and novel classes into the label space tests the bias of the fine-tuned model towards base classes. Hence, we believe this benchmark is a more realistic measure of the generalization performance of VLM fine-tuning methods. Though this setting has existed in traditional zero-shot learning (Xian et al., 2017), we introduce it back into the realm of VLM evaluation.

**(ii) Classification without Class-names (CwC).** VLMs require explicit class names to perform classification (Radford et al., 2021). This is a limitation for images whose label lies outside the VLM's vocabulary. CwC tests the ability of a VLM to classify truly novel images without explicitly using class names. During inference, all class names are replaced with the word *'object'*, and the model is tested on it's ability to classify an image based on descriptions alone. For example, to classify a *'Pikachu'* image, we just use the descriptions *{'has a yellow body', ..., 'has round red cheeks'}* and not the class name *'Pikachu'*, hence the text template looks like 'a photo of an object, which has a yellow body' etc. The model is fine-tuned on base classes, and evaluated on base and novel classes separately by removing all class names.

**Existing Evaluation Benchmarks:**

**(iii) Base-to-Novel Generalization (B2N).** In this setting, following prior work (Zhou et al., 2021; 2022; Khattak et al., 2022; Yao et al., 2023; Khattak et al., 2023; Shi & Yang, 2023), the dataset is split into equal disjoint base and novel classes, and the model is fine-tuned on few-shot (16-shot) training split of the base classes. During evaluation, unlike GZS, the label space is constrained to either just the base classes or just

| Dataset | | CLIP (ICML '21) | CoOp (IJCV '22) | VPT (ECCV '22) | CoCoOp (CVPR '22) | MaPLe (CVPR '23) | KgCoOp (CVPR '23) | ProGrad (ICCV '23) | PSRC (ICCV '23) | CLIP-VDT (ICCVW '23) | SAP (Ours) |
|---|---|---|---|---|---|---|---|---|---|---|---|
| **Average** | gBase | 60.81 | 75.19 | 73.48 | 73.13 | 75.47 | 76.86 | 70.15 | 78.81 | 63.75 | **79.47** (+0.66) |
| **on 11** | gNovel | 63.21 | 60.39 | 66.62 | 65.23 | 67.09 | 62.12 | 55.07 | 68.13 | 63.89 | **69.75** (+1.62) |
| **datasets** | gHM | 61.99 | 66.99 | 69.89 | 68.96 | 71.04 | 68.71 | 61.70 | 73.08 | 63.82 | **74.29** (+1.21) |

Table 2: Results on the GZS benchmark. gNovel & gBase indicate the accuracy of the novel classes and base classes respectively under the joint classification label space. gHM is the harmonic mean of gBase and gNovel. The best numbers are in **bold**, and the second best are underlined. SAP outperforms the best performing baseline on average gBase (by +0.66%), gNovel (by +1.62%), and gHM (by +1.21) computed across all datasets. Detailed dataset-wise results are presented in Appendix § D.

the novel classes. The testing phase for B2N is thus separate for base and novel classes, whereas the GZS benchmark has a unified testing phase.

**(iv) Cross-Dataset Generalization.** In this setting, the model is fine-tuned on ImageNet (Deng et al., 2009) and tested on the remaining datasets. This measures the ability of a VLM fine-tuning method to generalize to novel datasets.

**Baselines.** We compare SAP, against state-of-the-art baselines, including very recent prompt-tuning methods (summarized in Tab. 1), such as CLIP (Radford et al., 2021), CoOp (Zhou et al., 2021), VPT (Jia et al., 2022), CoCoOp (Zhou et al., 2022), ProDA (Lu et al., 2022), MaPLe (Khattak et al., 2022), KgCoOp (Yao et al., 2023), ProGrad (Zhu et al., 2023), PSRC (Khattak et al., 2023) and LoGoPrompt (Shi & Yang, 2023). We also compare against contemporary works that use external knowledge, such as KAPT (Kan et al., 2023), CLIP-VDT (Maniparambil et al., 2023) and CoPrompt (Roy & Etemad, 2024).

**Datasets.** We follow (Zhou et al., 2021; 2022; Khattak et al., 2022; 2023) to evaluate our method on 11 image classification datasets of varying complexity. These datasets encompass diverse domains, including generic object datasets like ImageNet (Deng et al., 2009) and Caltech101 (Fei-Fei et al., 2004); fine-grained datasets like Stanford Cars (Krause et al., 2013), OxfordPets (Parkhi et al., 2012), Flowers102 (Nilsback & Zisserman, 2008), Food101 (Bossard et al., 2014), FGVCAircraft (Maji et al., 2013); scene recognition dataset SUN397 (Xiao et al., 2010); action recognition dataset UCF101 (Soomro et al., 2012); texture dataset DTD (Cimpoi et al., 2013), and satellite image dataset EuroSAT (Helber et al., 2017).

**Overview of Results.** We present average base class accuracy, novel class accuracy, and their harmonic mean across 11 datasets for the GZS, CwC, B2N, and Cross-Dataset benchmarks in § 5.1 – Tab. 2, Fig. 4, Tab. 3, and Tab. 4 respectively. Dataset-wise expanded tables for all benchmarks, along with Domain Generalization and ResNet-50 backbone results are present in Appendix § D. In § 5.2, we show class activation maps to visualize image regions most relevant to a class description, where SAP demonstrates better localization capabilities. We study the goodness of our design choices in § 5.2 and show that part-level semantic alignment between image and text features helps learning better prompts.

## 5.1 Main Results

**(i) Generalized Zero-Shot Classification.** This newly proposed benchmark tests the ability of a method towards it bias to base classes and also it's generalization to novel classes within a dataset. We compare SAP against baselines and report the results in Tab. 2. The metric gBase is the average accuracy of test images belonging to base classes when the label space is the set of all classes (union of base and novel classes). The metric gNovel is the average accuracy of test images belonging to novel classes when the label space is the set of all classes. gHM is the harmonic mean of the gBase and gNovel. SAP's ability to leverage descriptions helps in mitigating the bias towards base classes, resulting in good generalized novel class accuracy. We outperform a recent state-of-the-art method PSRC, achieving better results in 8 out of 11 datasets (see Appendix § D), with a **+1.21%** margin in gHM averaged over all 11 datasets. Compared to the second-best method MaPLe, we have a significant margin of **+3.25%** in average gHM, outperforming it on all 11 datasets. We don't report the results of ProDA, LoGoPrompt, and KAPT in this setting due to code unavailability.

**(ii) Classification without Class-names** In this newly proposed benchmark, we study the ability of a pretrained VLM to classify images whose class names lie outside CLIP's vocabulary. Since the list of datasets

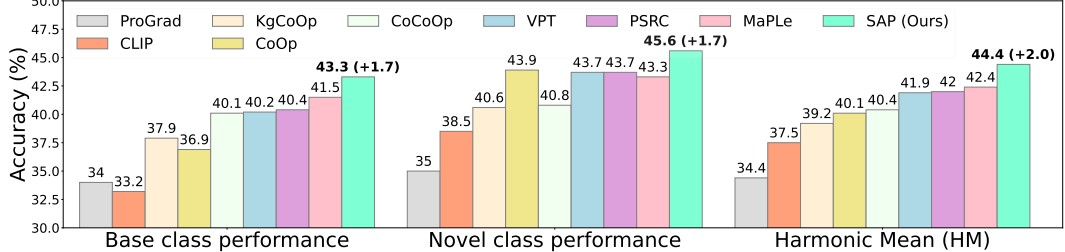

Figure 4: Comparison in the CwC setting. We show average Base, Novel, and HM accuracies over all 11 datasets. During evaluation, descriptions of each class are provided instead of the class name, and visual recognition is conducted based on these descriptions. SAP outperforms baselines by average Base (by +1.75%), Novel (by +1.76%) and HM (by +2.04%) computed over all datasets. Detailed dataset-wise results are presented in Appendix § D.

| Dataset | | CLIP | CoOp | VPT | CoCoOp | ProDA | MaPLe | KgCoOp | ProGrad | PSRC | L.Prompt | CLIP-VDT | KAPT | SAP (Ours) |
|---|---|---|---|---|---|---|---|---|---|---|---|---|---|---|
| Average on 11 datasets | Base | 69.34 | 82.69 | 80.81 | 80.47 | 81.56 | 82.28 | 80.73 | 82.48 | 84.26 | 84.47 | 82.48 | 81.10 | **84.68** (+0.21) |
| | Novel | 74.22 | 63.22 | 70.36 | 71.69 | 72.30 | 75.14 | 73.60 | 70.75 | 76.10 | 74.24 | 74.50 | 72.24 | **77.51** (+1.41) |
| | HM | 71.70 | 71.66 | 70.36 | 75.83 | 76.65 | 78.55 | 77.00 | 76.16 | 79.97 | 79.03 | 78.28 | 76.41 | **80.94** (+0.97) |

Table 3: Comparison on Base-to-Novel Generalization benchmark. The best numbers are in **bold**, and the second best are underlined. SAP outperforms the best performing baseline on average Base (by +0.21%), Novel (by +1.41%) and HM (by +0.97%) computed over all datasets. Expanded tables are in Appendix § D.

CLIP was trained on is not public knowledge, to empirically evaluate this setting we use the standard 11 datasets itself, but remove access to class-names during evaluation. Similar to the B2N setting, all models are trained on base-class images. For all baselines (including ours), we find the similarity of an image $\mathbf{x}$ with a class $y$ (not given to the model) as the average similarity between the image and the corresponding class-descriptions of $y$, which are known. We report average accuracies on 11 datasets in Fig. 4, where we outperform MaPLe (Khattak et al., 2022) by **+2.04%** in HM.

**(iii) Base-to-Novel Generalization.** We compare our method with twelve baselines and report the average accuracies in Tab. 3, where we outperform all baselines. We report per dataset accuracies in the Appendix § D, and show that SAP outperforms the state-of-the-art method PSRC in 7 out of 11 datasets while retaining performance in the others. We show significant gains in challenging datasets such as EuroSAT and DTD, where we outperform PSRC by a margin of **+5.66%** and **+2.92%** in HM respectively. We also show improved performance on the UCF-101 dataset, which contains a wide variety of human actions captured in diverse settings, where we show an improvement of **+2.49%** in HM over PSRC. These results indicate that SAP can integrate semantic knowledge provided by class descriptions to learn generalizable prompts.

**(iv) Cross-Dataset Generalization.** We compare our method with nine baselines and outperform all of them as shown in Tab. 4. SAP outperforms PSRC (Khattak et al., 2023) by **+1%** and MaPLe (Khattak et al., 2022) by **+0.5%** on average test accuracy over all datasets, which indicates that our method learns prompts that generalize across datasets.

| Dataset | CoOp | CoCoOp | VPT | MaPLe | KgCoOp | ProGrad | PSRC | CLIP-VDT | KAPT | SAP (Ours) |
|---|---|---|---|---|---|---|---|---|---|---|
| Avg. on 10 Datasets | 63.88 | 65.74 | 63.42 | 66.30 | 65.49 | 57.36 | 65.81 | 53.98 | 61.50 | **66.85** (+0.55) |

Table 4: Cross-Dataset Generalization. Models are trained on Imagenet and tested on the entire label space of new datasets without fine-tuning. SAP outperforms all baselines on average (see Appendix § D).

**Comparison against a recent method that uses external knowledge.** In Tab. 5 we compare SAP against CoPrompt (Roy & Etemad, 2024) on the B2N benchmark. Co-Prompt is a recent work that uses class descriptions to tune prompts and adapters, with a total of 4.74$M$ additional parameters over

| | | CoPrompt prompts+adapter | CoPrompt* prompts | SAP (Ours) prompts |
|---|---|---|---|---|
| Average on 11 datasets | Base | 84.00 | 83.40 | **84.68** (+1.28) |
| | Novel | 77.23 | 76.90 | **77.51** (+0.61) |
| | HM | 80.48 | 80.02 | **80.94** (+0.92) |

Table 5: B2N results comparison against CoPrompt. SAP outnumbers the prompt-only version by a margin on Base (by +1.28%), Novel (by +0.61%), and HM (by +0.92%).

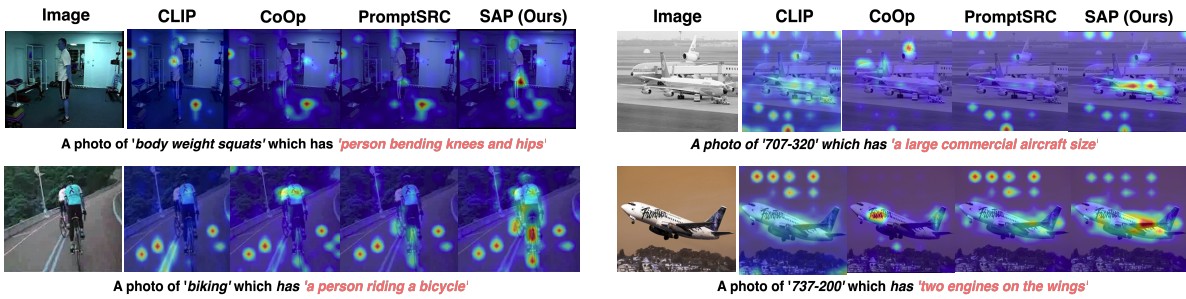

Figure 5: Images are highlighted at regions of highest activation relevant to specific text phrases, as identified by their prompted image and text encoders. Qualitatively, SAP localizes better than existing baselines.

CLIP. SAP outperforms CoPrompt by **+0.46%** average HM, despite only having $36K$ additional learnable parameters over CLIP. SAP outperforms a prompt-only version of CoPrompt, indicated by CoPrompt* in Tab. 5, by **+0.92%** in average HM.

## 5.2 Additional Results and Ablation Studies

**Class Activation Maps.** We present Class Activation Maps (CAMs) for the ViT-backbone CLIP image encoder to show image regions that most correlate to a given text description. We visualize activations of the pre-final self-attention layer of the transformer that maximize the cosine similarity between an image and a given text description. We present qualitative results in Fig. 5, where prompts learned by our method lead to better localizations. In § D, Tab. 17, we show quantitative results using an occlusion metric to measure the localization capabilities of our learned prompts.

**Study on Design Choices.** In this section we justify our design choice of *computing semantic alignment as the average similarity between the fused image feature and various description-guided text features*. Our key contribution is not just integrating descriptions into prompt learning for VLMs, but *how* descriptions are integrated into *both* visual and text modalities. We consider three alternative ways to incorporate class descriptions and show that our methodology leads to the best results. For our first alternative, we show that taking the *unnormalized mean* of description-guided text features to compute similarity leads to a drop in performance (SAP w/ mean text feature in Tab. 6). That is, computing semantic alignment as $\xi(\hat{\theta}_p(\mathbf{x}), \phi_p(y; A_y)) = sim(\hat{\theta}_p(\mathbf{x}), \frac{1}{|A_y|} \sum_{a \in A_y} \phi_p(y; a))$, leads to a drop in performance. This is in contrast to our design choice of taking the *mean* similarity, as shown in Eq. 1. Intuitively, descriptions of a class that are not well represented in pre-trained CLIP result in description-guided features with a low norm because CLIP has not encountered such associations during training. Information related to such descriptions is lost when the description-guided features are simply averaged out, without normalization.

We also observe that simply appending all class descriptions at once to generate a single description-guided text feature also leads to a drop in performance (SAP w/ agg descriptions in Tab. 6). Finally we show that replacing our text modality construction with that used by CLIP-VDT (CLIP-VDT text + SAP's Visual in Tab. 6) leads to a significant drop in average HM. These experiments show that how we add class descriptions is important, and that our approach is different from recent approaches that uses external information. We show average HM results across all 11 datasets of other design choices in Tab. 6.

| Method | Avg HM |
|---|---|
| SAP | **80.94** |
| SAP w/ mean text feature | 80.31 |
| SAP w/ agg descriptions | 79.17 |
| CLIP-VDT Text + SAP's Visual | 78.63 |

Table 6: Comparison with alternative design choices for incorporating class descriptions into the text modality.

**Effect of Removing Class Descriptions** Our method SAP incorporates class descriptions in both image and text modalities, as described in § 4.2 & § 4.3. Here we study the effect of removing description guidance from both modalities. To remove description guidance from the text modality, we just use the default class name template i.e. 'a photo of a [y]', without using any class description. We denote this baseline as SAP-TG. The results shown in Tab. 7 indicate that adding class descriptions to the text modality, as SAP does, results in a gain of +**1.53**% in average HM and a gain of +%**2.72** in avg Novel.

To study the effect of removing class descriptions from the image modality, we construct baselines by removing the cross attention module. We first consider a baseline that uses just the global image feature $\theta(x)$ instead of the fused feature and call this SAP w/ global. Then, we consider a baseline that naively combines global and local features (without incorporating class descriptions via cross-attention) by averaging them and denote it by SAP w/ global & local. We first compute the mean of the 196 local features to obtain a single mean local feature, and then average this mean local feature with the global feature. Note that both baselines construct description guided text features. The results presented in Tab. 7 justify our design choice of incorporating class descriptions into the image modality. Furthermore our method incorporates class descriptions into images through a fully non-parametric cross-attention, and adds no computational overhead.

| Method | Avg. Base | Avg. Novel | Avg. HM |
|---|---|---|---|
| Effect of Removing Class descriptions from the Text Modality | | | |
| SAP - TG | 84.62 | 74.79 | 79.41 |
| SAP | **84.68** | **77.51** | **80.94** |
| Effect of Removing Class Descriptions from the Image Modality | | | |
| SAP w/ global | 84.56 | 77.04 | 80.63 |
| SAP w/ global & local | 84.66 | 76.81 | 80.55 |
| SAP | **84.68** | **77.51** | **80.94** |

Table 7: All results are on the B2N generalization benchmark, and are average results over 11 datasets.

**Role of the Learnable Bias Term.** In this section we demonstrate the role of learnable bias in SAP. We add the bias term to enhance the learning of local image features. Poor local features arise because CLIP was originally trained using only the CLS token. Initial attempts to use the patch tokens directly to compute the local features resulted in poor performance due to pretrained layer norms (as in Fig 3). Therefore, we opt to adjust the local features in a parameter-efficient manner by adding a learnable bias. The bias term is necessary for obtaining high-quality local features. However, the bias alone is not sufficient; it must operate alongside cross-attention. The bias refines local features, while cross-attention integrates descriptive information into the visual domain, using local features. We study the effect of the bias along with our cross attention module in Tab. 8. The Class Descriptions column in Tab. 8. indicates their usage in the text modality. The table highlights that directly augmenting global image features with bias fine-tuned local features without cross-attention (row 2) performs worse than using just the global image features (row 4). This indicates that the bias alone is insufficient, and best results are obtained when the bias fine-tuned local features are integrated with class descriptions using our cross-attention module (row 1). Additionally, the result in row 3 suggest that without a learnable bias, the local features become uninformative, even when cross-attention is used. This indicates that the learnable bias plays a critical role in making the local features useful for semantic alignment. To further support the importance of the cross-attention module and bias together, we studied the performance drop when both the cross-attention and bias are removed across different architectures and settings. The results are shown in Tab. 9. The bias and cross-attention modules are complementary, and together they aid in incorporating class descriptions into the visual modality.

| Class Descriptions | Bias for Local features | Cross Attention | Avg. Novel | Avg. HM |
|---|---|---|---|---|
| ✓ | ✓ | ✓ | **77.51** | **80.94** |
| ✓ | ✓ | ✗ | 76.81 | 80.55 |
| ✓ | ✗ | ✓ | 75.72 | 79.9 |
| ✓ | ✗ | ✗ | 77.04 | 80.63 |

Table 8: Effect of learnable bias along with our cross-attention module on the B2N benchmark.

| Architecture-Setting | Avg. Novel | Avg. HM |
|---|---|---|
| ViT-B2N | -0.47 | -0.31 |
| RN50-B2N | -0.49 | -0.35 |
| ViT-GZS | -0.49 | -0.53 |
| RN50-GZS | -0.40 | -0.31 |

Table 9: Drop in the performance when both cross-attention and bias are removed from SAP across different architecture and settings.

## 6 Conclusions

Prompt learning has emerged as a valuable technique for fine-tuning VLMs for downstream tasks. However, existing methods encounter challenges such as overfitting due to limited training data and difficulties handling larger label spaces during evaluation, resulting in bias towards seen classes. Additionally, these methods struggle when class labels are not present in the vocabulary. We study if better text semantics can improve prompt learning, and propose an approach, named SAP, that learns prompts which better generalize to novel classes. Our proposed approach highlights that careful part-level semantic alignment between image and text features is crucial to leverage additional semantic information. We showcase the efficacy of our approach across four benchmarks, demonstrating significant improvements. We hope this work inspires further exploration into leveraging class descriptions in VLMs.

**Acknowledgements.** Hari Chandana Kuchibhotla would like to thank MoE for the PMRF fellowship support. Sai Srinivas Kancheti would like to thank MoE for the PMRF fellowship support, as well as Microsoft Research for the PhD Fellowship support. We thank the anonymous reviewers for their valuable feedback that improved the presentation of this paper.

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
