## Appendix

In this appendix, we present the following details.

- List of notations used in this paper and their descriptions are in § A.
- Overall algorithm of SAP is presented in § B.
- Implementation details are in § C.
- Expanded dataset-wise tables, and additional experiments are presented in § D.
- Examples of class descriptions generated using GPT-3.5 are presented in § E.
- Limitations and Broader Impact in § F.

## A  Summary of Notations and Terminology

We use $\cdot$ (*dot*) to represent various types of multiplication operations – matrix multiplication, matrix-vector or vector-matrix product, and vector dot-product. Detailed descriptions of notations are presented in Tab. 10.

| Notation | Description | Dimension |
|---|---|---|
| $\theta$ | Image Encoder | |
| $\phi$ | Text Encoder | |
| $\mathcal{Y}$ | Classification label space | |
| $\rho$ | Set of all learnable text and visual prompts | |
| $B$ | Batch size | |
| $N$ | Size of the set of descriptions | |
| $n$ | Number of the learnable prompt tokens | |
| $d$ | Dimension of the multimodal space | |
| $A_y$ | LLM generated descriptions for class $y$ | |
| $A$ | Union of all descriptions of the classification label space | |
| $\phi(A)$ | Class descriptions features | $\mathbb{R}^{N \times d}$ |
| $\phi(y; A_y)$ | Description-guided text features of class $y$ | $\mathbb{R}^{N \times d}$ |
| $\theta(x)$ | Global image feature | $\mathbb{R}^d$ |
| $\theta^l(x)$ | Local image feature | $\mathbb{R}^{M \times d}$ |
| $\theta^{desc}(x)$ | Description-guided image features | $\mathbb{R}^{N \times d}$ |
| $\bar{\theta}^{desc}(x)$ | Mean Description-guided image features | $\mathbb{R}^d$ |
| $\hat{\theta}(x)$ | Fused image features | $\mathbb{R}^d$ |
| $\theta_p(x)$ | Prompted Global image feature | $\mathbb{R}^d$ |
| $\theta_p^l(x)$ | Prompted Local image feature | $\mathbb{R}^{M \times d}$ |
| $\theta_p^{desc}(x)$ | Prompted Description-guided image features | $\mathbb{R}^{N \times d}$ |
| $\mathbf{r}$ | Description relevance score for an image | $\mathbb{R}^N$ |
| $\alpha$ | average specificity for all descriptions | $\mathbb{R}$ |

Table 10: Notations used in this paper and their descriptions.

## B  SAP: Algorithm

Algorithm 1 outlines the SAP methodology. The algorithm is summarized as follows: In a given dataset, descriptions for each class are acquired by querying the LLM (L1 - L4). Class description features are then derived by passing the descriptions through $\phi$ (L5). Unprompted and prompted image features are obtained by processing images through $\theta$ (L7-L8). The description-guided image features are obtained via a parameter-free cross-attention between local features and description features (L9). The local image features are a weighted average of the description-guided features based on the relevance of each description to the

image (L10 - L11). Finally, the mean description-guided image features and global image features are fused to create the fusion image feature (L12). Unprompted and prompted description-guided text features are obtained by passing the description-guided text templates through $\phi$ (L13-L14). $L_{ce}$, $L_{steer}^v$, and $L_{steer}^t$ loss functions are employed to train the prompts.

---

**Algorithm 1** SAP Algorithm

---

**Require:** Dataset D = $\{\mathbf{x}_i, y_i\}_{i=1}^B$; Classification label space: $\mathcal{Y}$; Vision and Language encoders: $(\theta, \phi)$; LLM: ChatGPT-3.5 model; Hyperparameters: coefficients $\lambda_1,\lambda_2$, scaling parameter $s$, learning rate $\delta$; Learnable Prompts: $\rho = \{\rho_t, \rho_v\}$
**Ensure:** Trained parameters $\hat{\rho}$
    */* Get descriptions for each class by querying LLM */*
1: **for** all y $\in \mathcal{Y}$ **do**
2:    $A_y$ = LLM(Visual features for distinguishing $y$ in a photo?)
3: **end for**
4: $A = \bigcup\limits_{y \in \mathcal{Y}} A_y$
5: $\phi(A)$ */* Get class description features */*
6: **for** all epochs **do**
6:    */* Get unprompted and prompted image features for every image $\mathbf{x}$ in the batch */*
7:    $\theta(\mathbf{x}), \_ = \theta(\mathbf{x})$
8:    $\theta_p(\mathbf{x}), \theta_p^l(\mathbf{x}) = \theta(\mathbf{x}; \rho_v)$
    */* Get description-guided image features using parameter-free cross-attention */*
9:    $\theta^{desc}(\mathbf{x}) = \text{Cross\_Attention}(Q = \phi(A), K = \theta^l(x), V = \theta^l(\mathbf{x}))$
    */* Get mean description-guided image feature using relevance score */*
10:    $\mathbf{r} = softmax(\phi(A) \cdot \theta(\mathbf{x}))$
11:    $\bar{\theta}^{desc}(\mathbf{x}) = \theta^{desc}(\mathbf{x})^\mathsf{T} \cdot \mathbf{r}$
    */* Get fused image feature by fusing global and local feature using description specificity ($\alpha$) */*
12:    $\hat{\theta}(\mathbf{x}) = (1 - \alpha) \cdot \theta(\mathbf{x}) + \alpha \cdot \bar{\theta}^{desc}(\mathbf{x})$
    */* Get unprompted and prompted description guided text features for every class $y$ */*
13:    $\phi(y, A_y) = \phi(y, A_y)$
14:    $\phi_p(y, A_y) = \phi(y, A_y; \rho_t)$
    */* Similarity between an image and a class is the aggregate of similarities over pertinent descriptions of a class */*
15:    $\xi(\hat{\theta}_p(\mathbf{x}), \phi_p(y; A_y)) = \frac{1}{|A_y|} \sum\limits_{a \in A_y} sim(\hat{\theta}_p(\mathbf{x}), \phi_p(y; a))$

16:    $L_{ce}(\rho) = -\frac{1}{B} \sum\limits_{i=1}^B \log \frac{\exp(\xi(\hat{\theta}_p(\mathbf{x_i}), \phi_p(y_i; A_{y_i}))/\tau)}{\sum\limits_{y \in \mathcal{Y}} \exp(\xi(\hat{\theta}_p(\mathbf{x_i}), \phi_p(y; A_y))/\tau)}$
    */* Compute Steering Losses */*

17:    $L_{steer}^v(\rho) = \frac{1}{B} \sum\limits_{i=1}^B \|\theta_p(\mathbf{x_i}) - \theta(\mathbf{x_i})\|_1$

18:    $L_{steer}^t(\rho) = \frac{1}{|\mathcal{Y}|} \sum\limits_{y \in \mathcal{Y}} \|\phi_p(y; A_y) - \phi(y; A_y)\|_1$
    */* Perform gradient descent on the total loss */*
19:    $\mathcal{L}(\rho) = L_{ce}(\rho) + \lambda_1 L_{steer}^v(\rho) + \lambda_2 L_{steer}^t(\rho)$
20:    $\hat{\rho} = \rho - \delta \nabla \mathcal{L}(\rho)$
21: **end for**
22: **return** $\hat{\rho}$

---

## C Implementation Details

**Training Details.** We use the ViT-B/16 (Dosovitskiy et al., 2021)-based CLIP model as our backbone. For the GZS and B2N benchmarks, we fine-tune the model on $K = 16$ shot training data from the base classes. Prompts are learned in the first three layers for the Cross-dataset benchmark and the first nine layers for the remaining two benchmarks. We introduce a $d$-dimensional bias as the sole additional parameter compared to (Khattak et al., 2023). The text prompts in the initial layer are initialized with the word embeddings of 'a photo of a', and the rest are randomly initialized from a normal distribution, similar to (Khattak et al., 2023). Our models are trained on a single Tesla V100 GPU with Nvidia driver version 470.199.02. We train for 20 epochs, with a batch size of 4 images, $\lambda_1 = 10$ and $\lambda_2 = 25$. The hyperparameter setup is common across all datasets. We use the SGD optimizer with a momentum of 0.9, a learning rate of 0.0025, and weight decay $5e - 4$. A cosine learning rate scheduler is applied with a warmup epoch of 1. We do not tune the temperature, and leave it at the default value of 100, also used by CLIP and PSRC. Image pre-processing involves random crops, random horizontal and vertical flips, and normalization using mean values of $[0.48, 0.46, 0.41]$ and standard deviation values of $[0.27, 0.26, 0.27]$. All baselines utilize publicly available codes and models. All results are averages over three seeds. We use PyTorch 1.12, CUDA 11.3, and build on the Dassl code repository: https://github.com/KaiyangZhou/Dassl.pytorch.

Our code is available at https://github.com/HariChandana1102/Semantic-Alignment-for-Prompt-Tuning-in-Vision-Language-Models

## D   Expanded Tables and Additional Results

**Using Random Text in place of Class Descriptions.** To study the usefulness of valid descriptions, we replace the descriptions for each class by randomly generated texting in Tab. 11. Examples of random descriptions are "Raindrops pattered softly against the roof", "A solitary figure walked down the empty street". We observe that descriptions matter for unusual datasets having texture-based images, satellite images, aircraft images and action recognition images. The average HM using random text across 11 datasets on B2N benchmark is **78.27%**, while SAP reports an average HM of **80.94%**. A drop of **2.67%** is noted.

|  | UCF101 | EuroSAT | DTD | OxfordPets | StanfordCars | Flowers102 | Food101 | FGVCAircraft | SUN397 | Caltech101 | ImageNet | Average |
|---|---|---|---|---|---|---|---|---|---|---|---|---|
| Base | 86.27 | 95.83 | 83.1 | 95.07 | 78.2 | 97.5 | 90.13 | 41.37 | 81.87 | 98.07 | 76.7 | 84.01 |
| Novel | 76.37 | 69.23 | 54.1 | 95.33 | 72.33 | 75.53 | 89.9 | 34.8 | 76.63 | 94.1 | 67.7 | 73.27 |
| HM | 81.02 | 80.39 | 65.54 | 95.2 | 75.15 | 85.12 | 90.01 | 37.8 | 79.16 | 96.04 | 72.17 | 78.27 |

Table 11: B2N benchmark results using random text in place of class descriptions. The results show that using irrelevant descriptions hurts model performance.

**Using Class Descriptions of Only Ground Truth Classes** Using class descriptions of the ground-truth class makes sense during training but may lead to noisy local features at inference. Our intention of using class descriptions of all *training classes*, is to construct a generalizable local view of the image, rather than a biased one. Due to the unbiased nature of the feature, it can help with tasks like Classification-without-Classnames. Tab. 12. shows the impact of using just the ground-truth class descriptions during training on three benchmarks. We do not change any hyperparameters. These results corroborate our perspective.

| B2N | Base | Novel | HM |
|---|---|---|---|
| all descriptions (Ours) | 84.68 | 77.51 | 80.94 |
| gnd truth descriptions | 84.58 | 76.93 | 80.58 |

| GZS | Base | Novel | HM |
|---|---|---|---|
| all descriptions (Ours) | 79.46 | 69.75 | 74.29 |
| gnd truth descriptions | 79.27 | 68.96 | 73.76 |

| CwC | Base | Novel | HM |
|---|---|---|---|
| all descriptions (Ours) | 43.30 | 45.60 | 44.40 |
| gnd truth descriptions | 41.76 | 43.45 | 42.59 |

Table 12: Comparison with ground truth class descriptions for B2N, GZS and CwC benchmarks.

**Using class descriptions from other LLMs.** We generate class descriptions from two other LLMs - OpenAI's GPT4o-mini OpenAI (2024) and Anthropic's Claude Haiku Anthropic (2024). Both LLMs considered are fast and cheap – for instance generating class descriptions for all classes of all 11 datasets from Claude Haiku takes 40 mins and costs 0.5$. The results are presented in the Tab. 13. for both B2N and GZS benchmarks:

| LLM | Base | Novel | HM |
|---|---|---|---|
| GPT-3.5 | 84.68 | 77.51 | 80.94 |
| Claude Haiku | 84.64 | 77.05 | 80.67 |
| GPT4o-mini | 84.74 | 77.16 | 80.77 |

| LLM | Base | Novel | HM |
|---|---|---|---|
| GPT-3.5 | 79.47 | 69.75 | 74.29 |
| Claude Haiku | 79.31 | 69.14 | 73.88 |
| GPT4o-mini | 79.54 | 69.53 | 74.2 |

Table 13: Comparison with class descriptions generated from other LLMs on Base-to-Novel benchmark on the left, and Generalized Zero-Shot benchmark on the right.

The results indicate that we get similar results across varying quality of outputs from different LLMs. We believe that in the future obtaining text semantics is going to be cheaper and easier, which necessitates algorithms that can make use of such cheap semantic information.

**Few-shot Setting.** Our main objective is to train prompts that can generalize effectively to novel classes and datasets. As such, we present results primarily on settings that test generalizability, such as the GZS benchmark, Base-to-Novel benchmark, and the Classification without Class-names benchmark. For completeness, we present results in a few-shot classification setting, where limited training samples are provided for all

classes. Note that there are no novel classes in this setting. We showcase outcomes for $K = 1, 2, 4, 8,$ and $16$ shots. As shown in Fig. 6, on average, across 11 datasets, we perform competitively against the best baseline PSRC.

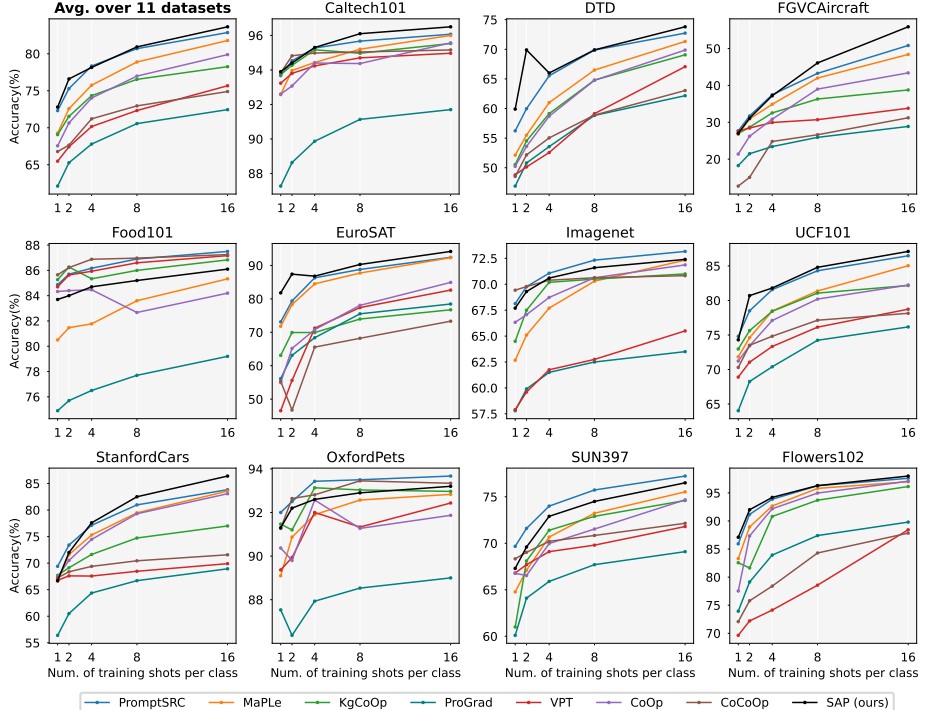

Figure 6: Performance of SAP in the few-shot setting. Our method achieves competitive performance compared to all baselines on average across 11 datasets.

**Domain Generalization.** We show results on Domain Generalization in Tab. 14. We train on $K = 16$ shot training data from base classes of source dataset ImageNet and evaluation on ImageNetV2, ImageNet-A, ImageNet-Setch, and ImageNet-R target datasets. SAP outperforms two strong baselines PSRC and MaPLe.

|  | Source | Target | | | | |
|---|---|---|---|---|---|---|
|  | ImageNet | -V2 | -A | -S | -R | Avg |
| MaPLe | 77.10 | 71.00 | 53.70 | 50.00 | 77.70 | 63.10 |
| PSRC | 76.30 | 71.00 | 54.10 | 50.00 | 77.80 | 63.22 |
| SAP | 76.40 | 71.10 | 55.70 | 49.80 | 77.50 | **63.52** |

Table 14: DG benchmark. SAP outperforms baselines on avg.

**ResNet-50 Backbone as Image Encoder.** Here we show the GZS and B2N performance of SAP using the ResNet-50 CLIP model as a backbone. We compare against five baselines which also use the ResNet-50 backbone and present our results in Tab. 15. For all methods including ours, we train the models without tuning any hyperparameters such as prompt-depth, regularization weight, learning rate etc. and use the same values as those of ViT-B/16 CLIP backbone. We observe that PSRC performs particularly poorly with a ResNet backbone. Although we use similar hyperparameters as PSRC, SAP shows good results, indicating that class descriptions help greatly in this setting. We show a gain of $\mathbf{+0.98\%}$ on average gHM for GZS, and $\mathbf{+2.32\%}$ on average HM in the B2N setting.

**Prompt Depth.** Tab. 16 shows the average HM for the B2N benchmark across nine datasets, excluding SUN397 and ImageNet. As seen from the table, adding prompts till depth 9 for image and text encoders is ideal for SAP performance and is used for B2N, GZS and CwC benchmarks.

| Depth | 1 | 3 | 5 | 7 | 9 | 11 |
|---|---|---|---|---|---|---|
| HM | 76.84 | 79.35 | 79.25 | 80.85 | **81.76** | 80.68 |

Table 16: Prompt depth analysis

**Class Activation Maps (CAMs).** We show additional CAMs for the ResNet-50(He et al., 2015) backbone encoder to visualize image regions that most correlate to a given description. Fig. 7 shows the GradCAM (Sel-

| Dataset | | CLIP | CoOp | KgCoOp | ProGrad | PSRC | SAP (Ours) |
|---|---|---|---|---|---|---|---|
| **Generalized Zero-Shot Learning Benchmark** | | | | | | | |
| **Average** | gBase | 57.01 | 68.65 | 69.25 | 69.89 | 47.41 | **71.52** (**+1.63**) |
| **on 11** | gNovel | **60.73** | 50.35 | 59.08 | 52.26 | 29.16 | 59.13 (**-1.60**) |
| **datasets** | gHM | 58.81 | 58.1 | 63.76 | 59.81 | 36.12 | **64.74** (**+0.98**) |
| **Base-to-Novel Generalization Benchmark** | | | | | | | |
| **Average** | Base | 65.27 | 77.24 | 75.51 | 77.98 | 55.13 | **78.49** (**+0.51**) |
| **on 11** | Novel | 68.14 | 57.40 | 67.53 | 63.41 | 38.72 | **69.32** (**+1.79**) |
| **datasets** | HM | 66.68 | 65.86 | 71.30 | 69.94 | 45.49 | **73.62** (**+2.32**) |

Table 15: Results on GZS and B2N settings using a ResNet-50 backbone. On average, SAP outperforms all the baselines.

varaju et al., 2017) visualizations for base classes *"Floor gymnastics"*, *"Hammering"*, *"Cape Flower"* and *"Highway"*. SAP effectively localizes the text semantics in the image compared to baselines. In Tab. 17, we show quantitative results using an occlusion metric to measure the localization capabilities of our learned prompts. Given a description, we mask out parts of the image which are most activated w.r.t. the description. The occluded image is then classified by the pre-trained CLIP model. A CAM localizes the description well if occluding image regions with the highest activations leads to a large drop in accuracy.

| Method | Archery | Baby Crawling | Band Marching | Apply Eye Makeup | Apply Lipstick | Biking | Body Weight Squats |
|---|---|---|---|---|---|---|---|
| CoOp | 57.39 | 64.42 | 61.99 | 75.00 | 78.66 | 55.15 | 53.97 |
| PSRC | 47.87 | 53.69 | 54.29 | 50.00 | 69.33 | 50.35 | 50.72 |
| Ours | **44.34** | **49.66** | **51.58** | **40.90** | **62.66** | **47.96** | **48.73** |
| | **707-320** | **747-200** | **737-200** | **727-200** | **C-130** | **CRJ-200** | **Boeing-717** |
| CoOp | 15.21 | 11.82 | 23.47 | 6.13 | 75.81 | 38.22 | 20.63 |
| PSRC | 6.14 | 8.84 | 21.42 | 3.06 | 75.86 | 32.45 | 23.58 |
| Ours | **3.00** | **5.92** | **15.30** | **0.00** | **60.61** | **26.58** | **14.72** |

Table 17: Occlusion benchmark (lower number is better): Images are masked at regions of highest activation relevant to a given class description, as identified by prompted image and text encoders, and then evaluated using the pre-trained CLIP model. The lower the accuracy, the better are the localizations. We show results for a few specific classes from the UCF101 dataset (top) and FGVC-Aircraft dataset (bottom). For example, for the class *'body weight squats'*, we use the description *'person bending knees and hips'*.

For instance, for the text phrase *'a photo of a 737-200, which has two engines on the wings'* we find that masking out important regions given by our prompted image encoder leads to an accuracy of 15.30%. This drop is higher than that of PSRC, whose accuracy drops only to 21.42%. This suggests that regions which are deemed important by SAP are highly correlated to the text phrase. Our parameter-free cross-attention module helps us learn prompts that focus on part-level image information.

**Expanded Dataset-wise Tables.** We present the elaborate tables dataset-wise for the Generalized Zero-Shot setting in Tab. 18 and Base-to-Novel generalization setting in Tab. 21. SAP outperforms the best-performing baseline, PSRC, in 7 of the 11 considered datasets. We perform very well in challenging datasets such as EuroSAT, DTD, and UCF-101. We present dataset-wise results for the Classification without Class-names benchmark in Tab. 19. Tab. 20 has the dataset-wise results for the Cross-Dataset generalizatin benchmark. In Tab. 15 we show average results on the GZS benchmark and the Base-to-Novel benchmark for the ResNet-50 backbone Image Encoder. We also present detailed, dataset-wise results for the same in Tab. 22.

# E   Generation of Class Descriptions

Tab. 23 shows class names sampled from different datasets and their respective descriptions retrieved using GPT-3.5 (Hagendorff et al., 2022). We use the query – "What are useful visual features for distinguishing a [classname] in a photo? Answer concisely." Class descriptions differ from well-

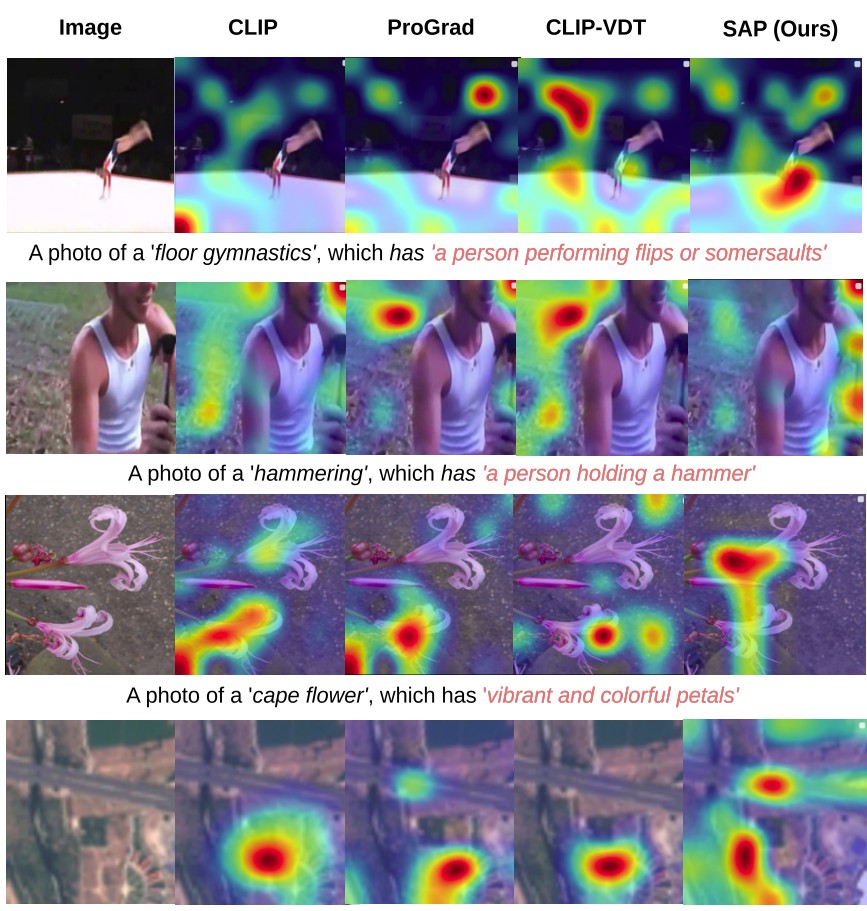

| Image | CLIP | ProGrad | CLIP-VDT | SAP (Ours) |

A photo of a '*floor gymnastics*', which *has* '*a person performing flips or somersaults*'

A photo of a '*hammering*', which *has* '*a person holding a hammer*'

A photo of a '*cape flower*', which has '*vibrant and colorful petals*'

A photo of a '*highway*', which is '*a long and straight path*'

Figure 7: Figure displays GradCAM visualizations that highlight the regions of highest activation relevant to specific text phrases. These visualizations use a ResNet-50 backbone as the image encoder for all baselines, including ours. SAP localizes better than the existing baselines.

curated attributes found in datasets with annotated attributes such as AwA (Lampert et al., 2009) and CUB (Wah et al., 2011) in three ways: (i) Our class descriptions may be noisy since no manual curation is used; (ii) They may not necessarily contain class-discriminative information, especially for similar classes; and (iii) Descriptions of a class are generated independently, and may not contain comparative traits w.r.t. other classes. These choices are primarily to keep our approach low-cost while integrating these finer details into fine-tuning of VLMs. It's important to note that our description generation occurs at the class level, not the image level, making it cost-efficient.

## F    Limitations and Broader Impact

A key dependency of our framework is the need for an LLM to provide descriptions at a class level. We however believe that this has become increasingly feasible in recent times, especially since we require at a class level and not at the image level. Our work deals with learning prompts for generalizable image classification by leveraging cheaply available semantic knowledge in the form of class descriptions. We believe that our work can serve as a stepping stone for incorporating semantic information to solve multi-modal tasks like captioning and VQA. To the best of our knowledge, there are no direct detrimental effects of our work.

| Dataset | | CLIP (ICML '21) | CoOp (IJCV '22) | VPT (ECCV '22) | CoCoOp (CVPR '22) | MaPLe (CVPR '23) | KgCoOp (CVPR '23) | ProGrad (ICCV '23) | PSRC (ICCV '23) | CLIP-VDT (ICCVW '23) | SAP (Ours) |
|---|---|---|---|---|---|---|---|---|---|---|---|
| **Average on 11 datasets** | gBase | 60.81 | 75.19 | 73.48 | 73.13 | 75.47 | 76.86 | 70.15 | 78.81 | 63.75 | **79.47** (+0.66) |
| | gNovel | 63.21 | 60.39 | 66.62 | 65.23 | 67.09 | 62.12 | 55.07 | 68.13 | 63.89 | **69.75** (+1.62) |
| | gHM | 61.99 | 66.99 | 69.89 | 68.96 | 71.04 | 68.71 | 61.70 | 73.08 | 63.82 | **74.29** (+1.21) |
| UCF101 | gBase | 62.70 | 80.26 | 75.76 | 76.56 | 76.90 | 78.96 | 74.63 | **82.67** | 66.19 | 82.23 |
| | gNovel | 64.40 | **84.76** | 67.73 | 64.76 | 70.40 | 62.33 | 51.36 | 71.40 | 67.00 | 76.40 |
| | gHM | 63.53 | **82.45** | 71.52 | 70.17 | 73.51 | 69.67 | 60.85 | 76.62 | 66.59 | 79.21 |
| EuroSAT | gBase | 51.40 | 69.26 | 88.22 | 70.86 | 84.06 | 82.02 | 76.26 | 86.60 | 55.09 | **94.37** |
| | gNovel | 38.90 | 36.26 | 53.36 | 41.03 | 43.90 | 31.26 | 23.43 | 54.16 | 50.79 | **58.53** |
| | gHM | 44.28 | 47.60 | 66.50 | 51.97 | 57.68 | 45.28 | 35.85 | 66.65 | 52.85 | **72.25** |
| DTD | gBase | 42.70 | 65.36 | 58.92 | 60.29 | 63.00 | 66.42 | 57.19 | **68.73** | 55.79 | 66.47 |
| | gNovel | 45.79 | 34.30 | 44.26 | 46.09 | 47.49 | 39.73 | 33.36 | 47.53 | 51.00 | **54.27** |
| | gHM | 44.19 | 44.99 | 50.55 | 52.25 | 54.16 | 49.72 | 42.14 | 56.20 | 53.28 | **59.75** |
| Oxford Pets | gBase | 84.80 | 89.56 | 89.06 | 91.12 | 91.69 | 91.99 | 88.36 | **93.00** | 83.80 | 91.97 |
| | gNovel | 90.19 | 90.46 | 93.23 | 92.50 | **93.93** | 92.69 | 87.76 | 91.00 | 90.40 | 92.30 |
| | gHM | 87.41 | 90.01 | 91.10 | 91.81 | **92.80** | 92.34 | 88.06 | 91.99 | 86.97 | 92.13 |
| Stanford Cars | gBase | 56.00 | 74.43 | 65.13 | 67.29 | 69.33 | 72.56 | 64.46 | 74.77 | 59.50 | **76.40** |
| | gNovel | 64.19 | 57.16 | 70.56 | 68.82 | 69.86 | 66.56 | 55.66 | **71.23** | 61.59 | 69.33 |
| | gHM | 59.81 | 64.67 | 67.74 | 68.05 | 69.61 | 69.43 | 59.74 | **72.96** | 60.52 | 72.69 |
| Flowers102 | gBase | 62.09 | 93.40 | 83.12 | 87.36 | 91.19 | 92.80 | 84.86 | 95.00 | 69.90 | **95.69** |
| | gNovel | 69.80 | 56.92 | 65.56 | 65.53 | 68.29 | 65.76 | 62.39 | 71.00 | 77.00 | **71.13** |
| | gHM | 65.71 | 70.74 | 73.31 | 74.89 | 78.10 | 76.97 | 71.92 | 81.27 | 73.20 | **81.60** |
| Food101 | gBase | 79.90 | 83.59 | 85.96 | 86.15 | 86.76 | 85.76 | 78.46 | **87.07** | 75.90 | 86.43 |
| | gNovel | 80.90 | 76.82 | 84.99 | 86.50 | **87.20** | 83.72 | 76.23 | 85.90 | 77.69 | 86.09 |
| | gHM | 80.39 | 80.07 | 85.49 | 86.33 | **86.98** | 84.73 | 77.33 | 86.48 | 76.78 | 86.26 |
| FGVC Aircraft | gBase | 14.50 | 29.92 | 25.12 | 25.90 | 25.90 | 32.69 | 23.93 | 34.90 | 16.10 | **35.00** |
| | gNovel | 23.79 | 22.83 | 28.03 | 26.36 | 28.53 | 22.06 | 15.63 | 28.40 | 18.60 | **30.23** |
| | gHM | 18.01 | 25.90 | 26.50 | 26.13 | 27.15 | 26.35 | 18.93 | 31.32 | 17.59 | **32.44** |
| SUN397 | gBase | 60.50 | 72.56 | 69.40 | 71.19 | 72.76 | 73.36 | 67.69 | **75.63** | 63.09 | 75.40 |
| | gNovel | 63.70 | 56.52 | 67.50 | 67.26 | 68.93 | 61.75 | 57.00 | 68.70 | 66.00 | **69.80** |
| | gHM | 62.05 | 63.55 | 68.44 | 69.17 | 70.79 | 67.06 | 61.89 | 72.00 | 64.51 | **72.30** |
| Caltech101 | gBase | 91.40 | 95.92 | 95.66 | 95.09 | 95.83 | 95.89 | 91.53 | 96.20 | 93.59 | **96.30** |
| | gNovel | 91.69 | 85.09 | 92.26 | 90.93 | 92.03 | 92.06 | 85.26 | 91.73 | 86.19 | **92.82** |
| | gHM | 91.54 | 90.19 | 93.94 | 92.97 | 93.89 | 93.94 | 88.29 | 93.91 | 89.73 | **94.53** |
| Imagenet | gBase | 63.00 | 72.80 | 71.9 | 72.59 | 72.80 | 73.00 | 64.19 | 72.30 | 61.79 | **73.97** |
| | gNovel | 62.00 | 63.20 | 65.40 | 67.80 | 67.40 | 65.40 | 57.70 | **68.40** | 56.59 | 66.66 |
| | gHM | 62.49 | 67.66 | 68.50 | 70.11 | 70.00 | 68.99 | 60.77 | **70.30** | 59.07 | 70.13 |

Table 18: Accuracy comparison on the GZS benchmark. gNovel & gBase indicate the accuracy of the novel classes and base classes respectively under the joint classification label space. gHM is the harmonic mean of gBase and gNovel. The best numbers are in bold, and the second best are underlined. As reported in the first row, SAP outperforms all baselines on average gBase (by +0.66%), gNovel (by +1.62%), and gHM (by 1.21%) computed across all datasets. We indicate the margin of improvement over the corresponding best-performing baseline for each metric in green.

| Dataset | | CLIP | CoOp | VPT | CoCoOp | MaPLe | KgCoOp | ProGrad | PSRC | SAP |
|---|---|---|---|---|---|---|---|---|---|---|
| **Average** | Base | 33.28 | 36.97 | 40.28 | 40.12 | _41.56_ | 37.95 | 34.00 | 40.40 | **43.31** (+1.75) |
| **on 11** | Novel | 38.55 | _43.90_ | 43.72 | 40.80 | 43.30 | 40.69 | 35.01 | 43.78 | **45.66** (+1.76) |
| **datasets** | HM | 35.72 | 40.14 | 41.93 | 40.46 | _42.41_ | 39.27 | 34.50 | 42.02 | **44.46** (+2.04) |
| UCF101 | Base | 56.60 | 61.20 | 61.20 | 61.70 | _64.20_ | 62.00 | 59.70 | 63.10 | **64.70** |
| | Novel | 62.20 | 66.80 | 63.20 | **70.70** | _70.40_ | 68.80 | 63.50 | 69.40 | 69.10 |
| | HM | 59.27 | 63.88 | 62.18 | 65.89 | **67.16** | 65.22 | 61.54 | 66.10 | _66.83_ |
| EuroSAT | Base | 39.90 | 47.10 | 76.50 | 62.90 | _84.30_ | 59.70 | 47.60 | 71.4 | **88.70** |
| | Novel | 71.10 | 78.70 | **83.20** | 49.00 | 58.30 | 57.60 | 45.80 | _82.10_ | 80.90 |
| | HM | 51.12 | 58.93 | _79.71_ | 55.09 | 68.93 | 58.63 | 46.68 | 76.38 | 84.62 |
| DTD | Base | 40.20 | 40.90 | _47.20_ | 44.20 | 44.90 | 41.90 | 39.20 | 42.70 | **52.40** |
| | Novel | 42.40 | 44.10 | 44.30 | _47.10_ | 42.90 | 44.40 | 40.20 | 44.00 | **49.00** |
| | HM | 41.27 | 42.44 | _45.70_ | 45.60 | 43.88 | 43.11 | 39.69 | 43.34 | **50.64** |
| Oxford Pets | Base | 24.50 | 32.00 | 22.30 | **34.20** | _32.80_ | 25.40 | 23.10 | 27.40 | 23.60 |
| | Novel | 35.20 | 40.80 | 40.70 | _44.10_ | **46.40** | 39.70 | 36.00 | 41.60 | _44.10_ |
| | HM | 28.89 | 35.87 | 28.81 | **38.52** | _38.43_ | 30.98 | 28.14 | 33.04 | 30.75 |
| Stanford Cars | Base | 13.50 | 15.60 | 17.60 | 16.30 | 10.30 | 12.50 | 10.00 | _21.00_ | **22.50** |
| | Novel | 15.90 | 20.70 | 18.90 | 11.70 | **25.80** | 15.30 | 8.50 | 20.40 | _23.40_ |
| | HM | 14.60 | 17.79 | 18.23 | 13.62 | 14.72 | 13.76 | 9.19 | _20.70_ | **22.94** |
| Flowers102 | Base | 7.40 | 14.10 | 12.40 | 17.70 | 18.30 | 12.00 | 16.40 | _18.80_ | **19.60** |
| | Novel | 9.30 | 20.40 | 18.40 | 17.60 | _23.20_ | 12.30 | 13.80 | 19.30 | **26.00** |
| | HM | 8.24 | 16.67 | 14.82 | 17.65 | _20.46_ | 12.15 | 14.99 | 19.05 | **22.35** |
| Food101 | Base | 35.10 | 42.70 | **44.00** | _43.40_ | 35.50 | 47.10 | 42.10 | 41.20 | 42.20 |
| | Novel | 33.80 | **45.40** | _44.80_ | 44.40 | 38.90 | 44.60 | 41.80 | 40.50 | 44.20 |
| | HM | 34.44 | _44.01_ | 44.40 | 43.89 | 37.12 | **45.82** | 41.95 | 40.85 | 43.18 |
| FGVC Aircraft | Base | 6.10 | _9.50_ | 8.00 | 7.00 | **13.40** | 6.80 | 5.20 | 8.30 | 9.40 |
| | Novel | 7.90 | **15.80** | 12.80 | 8.30 | _15.50_ | 10.70 | 8.20 | 12.30 | 12.30 |
| | HM | 6.88 | _11.87_ | 9.85 | 7.59 | **14.37** | 8.32 | 6.36 | 9.91 | 10.66 |
| SUN397 | Base | 46.60 | 49.20 | 50.50 | _51.30_ | 50.20 | 50.10 | 40.10 | 50.00 | **51.40** |
| | Novel | 48.30 | 50.00 | 51.40 | _52.50_ | 52.20 | **53.20** | 42.90 | 51.40 | 51.40 |
| | HM | 47.43 | 49.60 | 50.95 | **51.89** | 51.18 | _51.60_ | 41.45 | 50.69 | 51.40 |
| Caltech101 | Base | 77.80 | 76.00 | **83.00** | **83.00** | 82.30 | 80.80 | 72.30 | 81.10 | 81.70 |
| | Novel | 74.80 | 74.30 | _75.90_ | 75.80 | 75.50 | **76.20** | 63.20 | 75.10 | 75.20 |
| | HM | 76.27 | 75.14 | **79.29** | _79.24_ | 78.75 | 78.43 | 67.44 | 77.98 | 78.32 |
| ImageNet | Base | 18.40 | 18.40 | _20.40_ | 19.70 | **21.00** | 19.20 | 18.30 | 19.4 | 20.30 |
| | Novel | 23.20 | 26.00 | _27.40_ | **27.60** | 27.30 | 24.80 | 21.30 | 25.50 | 26.70 |
| | HM | 20.52 | 21.55 | _23.39_ | 22.99 | **23.74** | 21.64 | 19.69 | 22.04 | 23.06 |

Table 19: Accuracy comparison in the Classification without Class-names setting. We show average Base, Novel, and HM accuracies over all 11 datasets. During evaluation, descriptions of each class are provided instead of the class name, and visual recognition is conducted based on these descriptions. SAP outperforms baselines by average Base (by +1.75%), Novel (by +1.76%) and HM (by +2.04%) computed over all datasets.

| | Source | Target | | | | | | | | | | |
|---|---|---|---|---|---|---|---|---|---|---|---|---|
| | ImageNet | Caltech101 | OxfordPets | StanfordCars | Flowers102 | Food101 | Aircraft | SUN397 | DTD | EuroSAT | UCF101 | Average |
| CoOp | 71.51 | 93.70 | 89.14 | 64.51 | 68.71 | 85.30 | 18.47 | 64.15 | 41.92 | 46.39 | 66.55 | 63.88 |
| CoCoOp | 71.02 | 94.43 | 90.14 | 65.32 | 71.88 | 86.06 | 22.94 | 67.36 | 45.73 | 45.37 | 68.21 | 65.74 |
| VPT | 70.60 | 91.80 | 90.40 | 63.70 | 67.30 | 83.10 | 22.70 | 66.10 | 46.10 | 37.10 | 65.90 | 63.42 |
| MaPLe | 70.72 | 93.53 | 90.49 | 65.57 | 72.23 | 86.20 | 24.74 | 67.01 | 46.49 | 48.06 | 68.69 | 66.30 |
| KgCoOp | 69.94 | 94.08 | 90.13 | 65.63 | 71.21 | 86.48 | 23.85 | 67.47 | 45.80 | 41.98 | 68.33 | 65.49 |
| ProGrad | 62.17 | 88.30 | 86.43 | 55.61 | 62.69 | 76.76 | 15.76 | 60.16 | 39.48 | 28.47 | 58.70 | 57.36 |
| PSRC | 71.27 | 93.60 | 90.25 | 65.70 | 70.25 | 86.15 | 23.90 | 67.10 | 46.87 | 45.50 | 68.75 | 65.81 |
| CLIP-VDT | 68.10 | 85.40 | 83.50 | 50.30 | 56.00 | 72.50 | 14.60 | 56.30 | 42.70 | 24.70 | 53.80 | 53.98 |
| KAPT | N/A | 88.90 | 89.40 | 58.15 | 68.00 | 79.95 | 17.95 | N/A | 44.80 | 41.35 | 65.05 | 61.50 |
| SAP (Ours) | 71.40 | 94.53 | 90.14 | 64.58 | 71.31 | 86.23 | 24.47 | 68.09 | 48.61 | 49.10 | 71.52 | **66.85** |

Table 20: Cross-Dataset Generalization benchmark. Models are trained on Imagenet and tested on the entire label space of new datasets without fine-tuning. SAP outperforms all baselines on average. N/A: not available in (Kan et al., 2023).

| Dataset | | CLIP | CoOp | VPT | CoCoOp | ProDA | MaPLe | KgCoOp | ProGrad | PSRC | L.Prompt | CLIP-VDT | KAPT | SAP |
|---|---|---|---|---|---|---|---|---|---|---|---|---|---|---|
| **Average** | Base | 69.34 | 82.69 | 80.81 | 80.47 | 81.56 | 82.28 | 80.73 | 82.48 | 84.26 | 84.47 | 82.48 | 81.10 | **84.68** (+0.21) |
| **on 11** | Novel | 74.22 | 63.22 | 70.36 | 71.69 | 72.30 | 75.14 | 73.60 | 70.75 | 76.10 | 74.24 | 74.50 | 72.24 | **77.51** (+1.41) |
| **datasets** | HM | 71.70 | 71.66 | 70.36 | 75.83 | 76.65 | 78.55 | 77.00 | 76.16 | 79.97 | 79.03 | 78.28 | 76.41 | **80.94** (+0.97) |
| UCF101 | Base | 70.53 | 84.69 | 82.67 | 82.33 | 85.23 | 83.00 | 82.89 | 84.33 | **87.10** | 86.19 | 84.10 | 80.83 | 86.60 |
| | Novel | 77.50 | 56.05 | 74.54 | 77.64 | 78.04 | 80.77 | 76.67 | 76.94 | 78.80 | 73.07 | 76.40 | 67.10 | **83.90** |
| | HM | 73.85 | 67.46 | 78.39 | 77.64 | 78.04 | 80.77 | 79.65 | 79.35 | 82.74 | 79.09 | 80.07 | 73.33 | **85.23** |
| EuroSAT | Base | 56.48 | 92.19 | 93.01 | 87.49 | 83.90 | 94.07 | 85.64 | 90.11 | 92.90 | 93.67 | 88.50 | 84.80 | **96.10** |
| | Novel | 64.05 | 54.74 | 54.89 | 60.04 | 66.00 | 73.23 | 64.34 | 60.89 | 73.90 | 69.44 | 70.50 | 67.57 | **81.13** |
| | HM | 60.03 | 68.69 | 69.04 | 71.21 | 73.88 | 82.35 | 73.48 | 72.67 | 82.32 | 79.75 | 78.48 | 75.21 | **87.98** |
| DTD | Base | 53.24 | 79.44 | 79.15 | 77.01 | 80.67 | 80.36 | 77.55 | 77.35 | 83.37 | 82.87 | 81.80 | 75.97 | **84.27** |
| | Novel | 59.90 | 41.18 | 50.76 | 56.00 | 56.48 | 59.18 | 54.99 | 52.35 | 62.97 | 60.14 | 62.30 | 58.30 | **67.03** |
| | HM | 56.37 | 54.24 | 61.85 | 64.85 | 66.44 | 68.16 | 64.35 | 62.45 | 71.75 | 69.70 | 70.73 | 65.97 | **74.67** |
| Oxford Pets | Base | 91.17 | 93.67 | 94.81 | 95.20 | 95.43 | 95.43 | 94.65 | 95.07 | 95.33 | **96.07** | 94.40 | 93.13 | 95.27 |
| | Novel | 97.26 | 95.29 | 96.00 | 97.69 | **97.83** | 97.76 | 97.76 | 97.63 | 97.30 | 96.31 | 97.70 | 96.53 | 96.90 |
| | HM | 94.12 | 94.47 | 95.40 | 96.43 | **96.62** | 96.58 | 96.18 | 96.33 | 96.30 | 96.18 | 95.68 | 94.80 | 96.08 |
| Stanford Cars | Base | 63.37 | 78.12 | 72.46 | 70.49 | 74.70 | 72.94 | 71.76 | 77.68 | 78.27 | 78.36 | 76.80 | 69.47 | **79.70** |
| | Novel | 74.89 | 60.40 | 73.38 | 73.59 | 71.20 | 74.00 | **75.04** | 68.63 | 74.97 | 72.39 | 72.90 | 66.20 | 73.47 |
| | HM | 68.65 | 68.13 | 72.92 | 72.01 | 72.91 | 73.47 | 73.36 | 72.88 | **76.58** | 75.26 | 74.80 | 67.79 | 76.46 |
| Flowers102 | Base | 72.08 | 97.60 | 95.39 | 94.87 | 97.70 | 95.92 | 95.00 | 95.54 | 98.07 | **99.05** | 97.40 | 95.00 | 97.83 |
| | Novel | **77.80** | 59.67 | 73.87 | 71.75 | 68.68 | 72.46 | 74.73 | 71.87 | 76.50 | 76.52 | 75.30 | 71.20 | 76.50 |
| | HM | 74.83 | 74.06 | 83.26 | 81.71 | 80.66 | 82.56 | 83.65 | 82.03 | 85.95 | 86.34 | 84.94 | 81.40 | **86.86** |
| Food101 | Base | 90.10 | 88.33 | 89.88 | 90.70 | 90.30 | 90.71 | 90.50 | 90.37 | 90.67 | **90.82** | 90.40 | 86.13 | 90.40 |
| | Novel | 91.22 | 82.26 | 87.76 | 91.29 | 88.57 | **92.05** | 91.70 | 89.59 | 91.53 | 91.41 | 91.20 | 87.06 | 91.43 |
| | HM | 90.66 | 85.19 | 88.81 | 90.99 | 89.43 | **91.38** | 91.09 | 89.98 | 91.10 | 91.11 | 90.80 | 86.59 | 90.91 |
| FGVC Aircraft | Base | 27.19 | 40.44 | 33.10 | 33.41 | 36.90 | 37.44 | 36.21 | 40.54 | 42.73 | **45.98** | 37.80 | 29.67 | 42.93 |
| | Novel | 36.29 | 22.30 | 30.49 | 23.71 | 34.13 | 35.61 | 33.55 | 27.57 | 37.87 | 34.67 | 33.00 | 28.73 | **38.87** |
| | HM | 31.09 | 28.75 | 31.74 | 27.74 | 35.46 | 36.50 | 34.83 | 32.82 | 40.15 | 39.53 | 35.24 | 29.19 | **40.80** |
| SUN397 | Base | 69.36 | 80.60 | 79.66 | 79.74 | 78.67 | 80.82 | 80.29 | 81.26 | **82.67** | 81.20 | 81.40 | 79.40 | 82.57 |
| | Novel | 75.35 | 65.89 | 72.68 | 76.86 | 76.93 | 78.70 | 76.53 | 74.17 | 78.47 | 78.12 | 76.80 | 74.33 | **79.20** |
| | HM | 72.23 | 72.51 | 79.63 | 78.27 | 77.79 | 79.75 | 78.36 | 77.55 | 80.52 | 79.63 | 79.03 | 76.78 | **80.85** |
| Caltech101 | Base | 96.84 | 98.00 | 97.86 | 97.96 | 98.27 | 97.74 | 97.72 | 98.02 | 98.10 | 98.19 | **98.30** | 97.10 | 98.23 |
| | Novel | 94.00 | 89.91 | 93.76 | 93.81 | 93.23 | 94.36 | 94.39 | 93.89 | 94.03 | 93.78 | **95.90** | 93.53 | 94.37 |
| | HM | 95.40 | 93.73 | 95.77 | 95.84 | 95.68 | 96.02 | 96.03 | 95.91 | 96.02 | 95.93 | **97.09** | 95.28 | 96.26 |
| ImageNet | Base | 72.43 | 76.47 | 70.93 | 75.98 | 75.40 | 76.66 | 75.83 | 77.02 | **77.60** | 76.74 | 76.40 | 71.10 | **77.60** |
| | Novel | 68.14 | 67.88 | 65.90 | 70.43 | 70.23 | 70.54 | 69.96 | 66.66 | 70.73 | **70.83** | 68.30 | 65.20 | 69.83 |
| | HM | 70.22 | 71.92 | 73.66 | 73.10 | 72.72 | 73.47 | 72.78 | 71.46 | **74.01** | 73.66 | 72.12 | 68.02 | 73.51 |

Table 21: Accuracy comparison on Base-to-Novel Generalization benchmark. The best numbers are in bold, and the second best are underlined. SAP outperforms all baselines on average Base (by +0.21%), Novel (by +1.41%) and HM (by +0.97%) computed over all datasets. We indicate the margin of improvement over the corresponding best-performing baseline for each metric in green.

| | | GZS Benchmark | | | | | | | Base-to-Novel Benchmark | | | | | |
|---|---|---|---|---|---|---|---|---|---|---|---|---|---|---|
| Dataset | | CLIP | CoOp | KgCoOp | Pro-Grad | PSRC | SAP | | CLIP | CoOp | KgCoOp | Pro-Grad | PSRC | SAP |
| **Average** | gBase | 57.01 | 68.65 | 69.25 | 69.89 | 47.41 | **71.52** (+1.63) | Base | 65.27 | 77.24 | 75.51 | 77.98 | 55.13 | **78.49** (+0.51) |
| **on 11** | gNovel | 60.73 | 50.35 | 59.08 | 52.26 | 29.16 | 59.13 (-1.60) | Novel | 68.14 | 57.40 | 67.53 | 63.41 | 38.72 | **69.32** (+1.79) |
| **datasets** | gHM | 58.81 | 58.10 | 63.76 | 59.81 | 36.12 | **64.74** (+0.98) | HM | 66.68 | 65.86 | 71.30 | 69.94 | 45.49 | **73.62** (+2.32) |
| UCF101 | gBase | 61.20 | 73.20 | 71.05 | 72.75 | 51.55 | **74.73** | Base | 68.40 | 79.78 | 77.16 | **81.04** | 59.95 | 80.70 |
| | gNovel | 61.79 | 45.10 | 56.95 | 48.05 | 30.25 | **63.80** | Novel | 61.50 | 48.31 | 70.13 | 60.07 | 38.85 | **72.67** |
| | gHM | 61.49 | 55.81 | 63.22 | 57.87 | 38.13 | 68.33 | HM | 64.77 | 60.18 | 73.48 | 69.00 | 47.15 | **76.47** |
| EuroSAT | gBase | 32.79 | 62.70 | 71.25 | **73.60** | 61.15 | 72.77 | Base | 55.80 | 90.25 | 84.28 | 88.44 | 70.35 | **91.33** |
| | gNovel | **46.50** | 23.45 | 33.95 | 19.40 | 09.00 | 32.32 | Novel | 66.90 | 31.30 | 53.53 | 49.49 | 33.90 | **67.00** |
| | gHM | 38.46 | 34.13 | **45.99** | 30.71 | 15.69 | 44.76 | HM | 60.85 | 46.48 | 65.47 | 63.47 | 45.75 | **77.30** |
| DTD | gBase | 43.50 | 60.60 | 64.80 | 62.30 | 42.60 | 62.57 | Base | 53.70 | 75.12 | 74.73 | 73.80 | 51.35 | **75.97** |
| | gNovel | 41.29 | 27.05 | 40.45 | 27.05 | 18.30 | **44.27** | Novel | 55.60 | 37.08 | 48.39 | 46.38 | 29.85 | **57.90** |
| | gHM | 42.37 | 37.40 | 49.81 | 37.72 | 25.60 | **51.91** | HM | 54.63 | 49.65 | 58.74 | 56.96 | 37.75 | **65.72** |
| Oxford Pets | gBase | 85.90 | 84.70 | 85.75 | 85.95 | 67.65 | **87.00** | Base | 91.20 | 90.15 | 92.57 | 92.36 | 77.60 | 91.90 |
| | gNovel | 85.59 | 85.25 | **90.45** | 87.10 | 65.65 | 89.27 | Novel | 93.90 | 90.70 | **94.61** | 94.48 | 79.40 | 94.57 |
| | gHM | 85.74 | 84.97 | 88.04 | 86.52 | 66.63 | **88.12** | HM | 92.53 | 90.42 | 93.58 | 93.41 | 78.49 | 93.22 |
| Stanford Cars | gBase | 48.29 | 64.70 | 62.25 | 64.30 | 17.35 | **68.20** | Base | 55.50 | 68.89 | 63.28 | **71.79** | 26.35 | 71.43 |
| | gNovel | **64.09** | 48.05 | 59.20 | 53.45 | 21.65 | 57.60 | Novel | 66.50 | 57.13 | **66.92** | 59.36 | 25.50 | 64.77 |
| | gHM | 55.08 | 55.15 | 60.69 | 58.38 | 19.26 | **62.45** | HM | 60.50 | 62.46 | 65.05 | 64.99 | 25.92 | **67.94** |
| Flowers102 | gBase | 62.59 | 89.40 | 85.70 | 88.80 | 65.00 | **92.52** | Base | 69.70 | 95.22 | 91.45 | 94.71 | 73.75 | **96.40** |
| | gNovel | **68.30** | 50.70 | 63.85 | 52.75 | 10.85 | 61.62 | Novel | 73.90 | 59.53 | **71.75** | 68.86 | 19.75 | 70.30 |
| | gHM | 65.32 | 64.70 | 73.18 | 66.18 | 18.60 | **73.97** | HM | 71.74 | 73.26 | 80.41 | 79.74 | 31.16 | **81.31** |
| Food101 | gBase | 75.80 | 73.80 | **78.30** | 76.30 | 32.65 | 77.97 | Base | 83.10 | 81.70 | 83.90 | 83.77 | 37.85 | 83.57 |
| | gNovel | **78.90** | 68.50 | 78.25 | 72.90 | 17.60 | 76.60 | Novel | 84.50 | 78.13 | **85.23** | 83.74 | 27.15 | 84.13 |
| | gHM | 77.32 | 71.05 | **78.27** | 74.56 | 22.87 | 77.28 | HM | 83.79 | 79.88 | **84.56** | 83.75 | 31.62 | 83.85 |
| FGVC Aircraft | gBase | 12.69 | **24.15** | 20.20 | 21.60 | 8.65 | 23.17 | Base | 18.80 | 28.39 | 24.91 | **30.17** | 14.20 | 28.97 |
| | gNovel | 22.10 | 14.75 | **18.20** | 14.25 | 6.95 | 17.45 | Novel | 26.00 | 20.02 | **25.69** | 19.70 | 9.05 | 25.33 |
| | gHM | 16.12 | 18.31 | 19.15 | 17.17 | 7.71 | **19.91** | HM | 21.82 | 23.48 | 25.29 | 23.84 | 11.05 | **27.03** |
| SUN397 | gBase | 56.70 | 66.65 | 67.05 | 67.15 | 54.25 | **70.40** | Base | 66.40 | 76.33 | 75.33 | 76.90 | 63.25 | **78.20** |
| | gNovel | 60.50 | 53.30 | 61.80 | 56.50 | 45.85 | **62.20** | Novel | 70.10 | 62.89 | 72.25 | 68.09 | 57.50 | **73.27** |
| | gHM | 58.54 | 59.23 | 64.32 | 61.37 | 49.70 | **66.05** | HM | 70.10 | 68.96 | 73.76 | 72.23 | 60.24 | **75.65** |
| Caltech101 | gBase | 88.59 | 91.35 | 91.65 | 91.50 | 79.35 | **92.13** | Base | 91.00 | 95.20 | 95.35 | **95.72** | 84.80 | 95.67 |
| | gNovel | 81.69 | 82.15 | **88.05** | 86.30 | 58.65 | 87.50 | Novel | 90.60 | 87.55 | **91.92** | 89.92 | 65.65 | 91.13 |
| | gHM | 85.00 | 86.51 | **89.81** | 88.82 | 67.45 | 89.76 | HM | 90.80 | 91.21 | **93.60** | 92.73 | 74.01 | 93.34 |
| ImageNet | gBase | 59.09 | 63.90 | 63.75 | 64.55 | 41.40 | **65.05** | Base | 64.40 | 68.5 | 67.67 | 69.13 | 47.00 | **69.20** |
| | gNovel | 57.29 | 55.60 | **58.75** | 57.15 | 36.05 | 57.85 | Novel | 60.10 | 58.76 | **62.45** | 57.39 | 39.35 | 61.40 |
| | gHM | 58.18 | 59.46 | 61.15 | 60.63 | 38.54 | **61.24** | HM | 62.18 | 63.29 | 64.96 | 62.72 | 42.84 | **65.07** |

Table 22: GZS benchmark and Base-to-Novel Generalization benchmark using ResNet backbone. Metrics for the GZS benchmark, such as gBase, gNovel, and gHM, are employed in the left section of the table. Conversely, metrics like Base, Novel, and HM are utilized to assess the Base-to-Novel benchmark in the right section. On average, our method outperforms all the baselines.

| Class (Dataset) | Descriptions | Class (Dataset) | Descriptions |
|---|---|---|---|
| Breast stroke (UCF101) | 1. Arms moving in a circular motion
2. Kicking legs in a frog-like motion
3. Head above water during stroke
4. Positioned horizontally in the water
5. Pushing water forward and outwards | Diving (UCF101) | 1. Person in mid-air or jumping
2. Person wearing diving gear
3. water splashing or ripples
4. Person wearing gogglesr
5. Person wearing swim cap |
| Highway or road (EuroSAT) | 1. Long and straight path
2. Multiple lanes for traffic
3. Traffic signs
4. Smooth and paved surface
5. Guardrails or barriers | Permanent cropland (EuroSAT) | 1. Uniform vegetation or crops
2. Irrigation systems or canals
3. Organized rows or patterns
4. Fences or boundaries
5. Distinct crop types or varieties |
| Striped (DTD) | 1. Alternating bands or lines
2. Regular pattern of stripes
3. Varying widths of stripes
4. Contrasting colors between stripes
5. Horizontal, vertical, diagonal stripes | Wrinkled (DTD) | 1. Irregular and uneven surface
2. Creases or folds
3. Shadows indicating unevenness
4. Lack of smoothness
5. Distorted or crumpled appearance |
| Maine coon (Oxford Pets) | 1. Large domestic cat
2. Long, bushy tail
3. Tufted ears with lynx-like tips
4. Rectangular body shape
5. Tufted paws | Chihuahua (Oxford Pets) | 1. Small breed of dog
2. Rounded apple-shaped head
3. Erect, pointy ears
4. Short snout
5. Short legs and long tail |
| 2008 chrysler pt cruiser convertible (Stanford Cars) | 1. Convertible top
2. Chrome grille
3. PT cruiser badge
4. Alloy wheels
5. Boxy shape | 2012 ferrari ff coupe (Stanford Cars) | 1. Sleek and sporty design
2. Large and stylish alloy wheels
3. Low and wide stance
4. Ferrari logo on the front and rear
5. Dual exhaust pipes |
| Watercress (Flowers102) | 1. Small, round-shaped leaves
2. Vibrant green color
3. Thin, delicate stems
4. Water or moist environments
5. Clusters of small white flowers | Trumpet creeper (Flowers102) | 1. Bright orange or red flowers
2. Trumpet-shaped blossoms
3. Long, tubular petals
4. Green leaves with serrated edges
5. Hummingbirds and bees |
| Hot dog (Food101) | 1. Cylindrical-shaped food
2. Bun or bread
3. Sausage or frankfurter
4. Visible grill marks
5. Toppings like onions or relish | Sushi (Food101) | 1. Bite-sized and compact
2. Rice as a base
3. Raw or cooked fish
4. Seaweed wrapping (nori)
5. Served with soy sauce |
| 737-200 (FGVC Aircraft) | 1. Two engines on the wings
2. Low wing configuration
3. Narrow body
4. Distinctive short fuselage
5. Swept-back wings | Industrial area (SUN397) | 1. Factories or warehouses
2. Smokestacks or chimneys
3. Cranes or heavy machinery
4. Conveyor belts or assembly lines
5. Trucks or shipping containers |
| Gramophone (Caltech101) | 1. Phonograph Cylinder or Disc
2. Horn Speaker
3. Hand-Cranked Operation
4. Nostalgic and Vintage Appeal
5. Vinyl or Shellac Records | Buckle (Imagenet) | 1. Metal or plastic object
2. Rectangular or circular shape
3. Fastening or securing
4. Opened and closed
5. Found on belts or straps |

Table 23: Sample classes from various datasets and the corresponding descriptions provided by GPT-3.5.