# OpenReview forum: "Semantic Alignment for Prompt-Tuning in Vision Language Models"
_TMLR — Accepted by TMLR_

### Review · Reviewer_3NEX · 2024-08-05

**Summary Of Contributions:**

The authors introduced a method for prompt-tuning in VLMs called Semantic Alignment for Prompt-tuning (SAP). The primary contribution is leveraging class descriptions generated by LLMs to enhance the semantic alignment between image and text modalities at a part level. This approach addresses overfitting in low-shot training scenarios and improves generalization to unseen classes and datasets. The authors propose new evaluation protocols, Generalized Zero-Shot Classification, and Out-of-Vocabulary Classification, demonstrating that SAP significantly outperforms existing state-of-the-art methods across multiple benchmarks. The method's efficiency and effectiveness are validated through comprehensive experiments on 11 benchmark datasets.

**Audience:**

Yes

**Claims And Evidence:**

Yes

**Requested Changes:**

Please try to address the comments on weakness.

**Strengths And Weaknesses:**

Strengths:
- The SAP method's focus on part-level semantic alignment between image and text features is interesting, demonstrating improved performance and localization capabilities.
- The introduction of the Generalized Zero-Shot Classification and Out-of-Vocabulary Classification benchmarks are also useful.
- The authors conduct extensive experiments across 11 benchmark datasets, proving the method's improvements to existing approaches.

Weaknesses:
- The method heavily relies on the quality of class descriptions generated by LLMs. Any noise or inaccuracies in these descriptions could impact performance. It is better to discuss the impacts of these noises or inaccuracies for responses from the LLMs.

- The method involves multiple sophisticated steps such as cross-attention mechanisms and fusion of global and local features, which might be complex to implement and replicate. How about the efficiency of using the proposed SAP when compared with other previous methods?

- The differences in performance across different settings in the Effect of Removing Class Descriptions are not significant. We observe only minor changes in the metrics for different settings. The paper may also need to provide some hints about tuning the temperature parameter, which may be quite important for the final performance.

---

> ### Author Response · Authors · 2024-09-07
> **Response to Reviewer 3NEX**
>
> Thank you for your insightful feedback. We have responded to each of your comments/suggestions below. We have also summarized our core contributions in this work in the common response to all reviewers.
>
> > 1. The method heavily relies on the quality of class descriptions generated by LLMs. Any noise or inaccuracies in these descriptions could impact performance. It is better to discuss the impacts of these noises or inaccuracies for responses from the LLMs.
>
> There are two major causes of noise in the class descriptions.
>
> 1. As explained in Section 4.1, we *generate class descriptions at a class name level, not for individual images*.  This could lead to scenarios where some descriptions are not precise for a specific image. For instance, an image of a front-facing cat may have its tail occluded, and the class description "has a long tail" is not entirely relevant for this particular cat image.
> Our formulation handles this noise via a _relevance score_ as discussed in Section 4.3 (equation 3 in the revised version).
>
> 2. LLMs can inherently be noisy. Our results in the paper show that our approach gives state-of-the-art results even with such noisy text descriptions. We do not perform any postprocessing on the generated descriptions. To study this further, we considered other LLMs (beyond what we had in our earlier version), and find that the impact of the specific LLM used is negligible.
> In particular, we generated class descriptions from two other LLMs - OpenAI's GPT4o-mini and Anthropic's Claude Haiku. Both LLMs considered are fast and cheap -- for instance, generating class descriptions from Claude Haiku takes ~40 mins and costs $0.5 for all 11 datasets and 4000 class labels considered across our experiments. Our results are presented in the table below:
>
> Generalized Zero-Shot Benchmark
> | LLM | Avg Base Acc | Avg Novel Acc | Avg HM |
> | ---- | ---------- | --------------| --------|
> |GPT-3.5 | 79.47 |  69.75| 74.29 |
> | Claude Haiku | 79.31 | 69.14 | 73.88 |
> | GPT4o-mini |79.54 |69.53  | 74.2|
>
> Base-to-Novel Generalization Benchmark
> | LLM | Avg Base Acc | Avg Novel Acc | Avg HM |
> | ---- | ---------- | --------------| --------|
> |GPT-3.5 | 84.68 | 77.51 | 80.94 |
> | Claude Haiku | 84.64 | 77.05 | 80.67 |
> | GPT4o-mini |84.74 | 77.16 | 80.77|
>
> We have added these results to Section D of the Appendix in the revised manuscript. The results indicate that we get similar results across varying quality of outputs from different LLMs; we however acknowledge that the LLM must be one that is capable of generating informative descriptions. As discussed further in our response to Reviewer uP8W, we believe that access to strong LLMs may only improve in the near future, making this approach feasible.
>
> We also study how uninformative descriptions may impact the performance of our method. We replace class decriptions with random words, and we find that this leads to a significant drop in performance as shown in Table 10 of Appendix Section D. We hence infer that a reasonable level of text semantics obtained from class labels can be very useful for learning prompt tuning methods in adaptation of VLMs for classification.
>
>
> > 2. The method involves multiple sophisticated steps such as cross-attention mechanisms and fusion of global and local features, which might be complex to implement and replicate. How about the efficiency of using the proposed SAP when compared with other previous methods?
>
> The **cross-attention** mechanism is a simple and popular technique extensively used in literature. We chose a parameter-free variant of cross-attention which does not add any computational overhead to the method. As described in Sec 4.3, the **fusion component** is also lightweight and is a simple heuristic to combine global and local features. Neither component involves any learnable parameters, and does not add any overhead. Our method, SAP's inference time is the same as PromptSRC while, in fact, being slightly faster than MaPLe. Our method is fairly straightforward to implement, and we plan to make our code publicly available for reproducibility and further research.

---

> > ### Author Response · Authors · 2024-09-07
> > **[Contd] Response to Reviewer 3NEX**
> >
> > > 3. The differences in performance across different settings in the Effect of Removing Class Descriptions are not significant. We observe only minor changes in the metrics for different settings. The paper may also need to provide some hints about tuning the temperature parameter, which may be quite important for the final performance.
> >
> > We beg to respectfully disagree. It is common for methods in this space to show upto 1% improvement over the previous state-of-the-art; for example, on average, PromptSRC showed a gain of 0.96% over MaPLe for novel classes.
> > The table below summarizes the empirical gains SAP achieves over PromptSRC (PSRC) in a variety of settings. In the Base2new setting, SAP shows a gain of 1.41% over PSRC for novel classes (which measures ability to generalize to new classes). Evidently, these improvements are similar or better than improvements obtained by earlier methods in the field.
> >
> >
> > | Evaluation    | PSRC  | SAP   | SAP's Improvement |
> > | ------------- | ----- | ----- | ----------------- |
> > | GZS  (Novel)   | 68.13 | 69.75 | **+1.62**         |
> > | OVC (Novel)    | 43.70 | 45.60 | **+1.90**         |
> > | B2N   (Novel)  | 76.10 | 77.51 | **+1.41**         |
> > | Cross-Dataset | 65.81 | 66.85 | **+1.04**         |
> > | Domain Generalization              |   63.22    |   63.52    | **+0.30**                  |
> > |GZS (Novel) (ResNet Backbone)|29.16|59.13|**+29.97**|
> > |B2N (Novel) (ResNet Backbone)|38.72|69.32|**+30.60**|
> >
> >
> >
> > In the implementation details section of the paper (Appendix Sec C), we provide a comprehensive description of the hyperparameters used. We use the same configuration for all datasets. **We do not** tune the temperature parameter, and leave it at the default value of 100, also used by CLIP and PromptSRC. We add this detail to Appendix Sec C. For fair comparison with baseline methods, we avoid detailed hyperparameter tuning techniques or mixed precision training; the improvements presented in our results come directly as a consequence of our methodological contributions in using only class descriptions and object-level features to perform semantic alignment and not from any other hyperparameter choices beyond the baselines. (We follow the hyperparameters of the strongest existing baseline PromptSRC.)
> >
> > Your suggestions have greatly improved the presentation of our paper. We appreciate this opportunity to respond to your queries, and will be happy to answer further as required.

---

### Review · Reviewer_xQ13 · 2024-08-12

**Summary Of Contributions:**

The paper proposes a strategy to perform soft prompt-tuning while finetuning a contrastive VLM. It has two major contributions:

* Replace standard CLIP style prompts “a photo of a {}” with class descriptors as obtained with GPT-3.
* Cross-attention module between the class descriptor and local image representations.

The paper evaluates the proposed technique on Cross-Dataset Generalization and Base-to-Novel Generalization. They propose two further tasks: Generalized Zero-Shot Classification and Out-of-Vocabulary Classification.

* The difference between Generalized Zero-Shot classification and Base-to-Novel generalization is that the label space is shared between base and novel classes at inference time.
* The Out-of-Vocabulary classification benchmark replaces the class names with descriptors at test time.

Qualitative analysis is given on GRAD-CAM based visualization. Ablations are provided decoupling the importance of various components in the model.

**Audience:**

Yes

**Broader Impact Concerns:**

The paper performs VLM finetuning on popular benchmarks. Hence, there are no broader impact concerns.

**Claims And Evidence:**

No

**Requested Changes:**

### Crucial

* The major claim of the paper is that “semantic alignment” between the class descriptors and the local image features is crucial to achieve good performance. However, Table 8 displays that the impact of removing the cross-attention module is minimal to the final performance (+0.3-0.4). The improvement that comes from just incorporating the learnable bias parameter (+1.0) is much higher than the cross-attention module. Thus, the paper may adjust this strong claim including the introduction, abstract and title and also emphasize the importance of this learnable bias.
* Further details of how hyperparameters $\lambda_1$, $\lambda_2$ and the number of soft prompts are tuned may be reported. This could be contrasted with the different baseline methods.
* How is $\alpha$ chosen?
* While finetuning the VLM, the class-descriptions of the ground-truth class are readily available. So, to compute $\theta_{desc}$, one can just use the class-decriptions of the ground-truth class instead of using all class descriptions $A \in R^{N \times D}$. Using all class descriptions is quite noisy and also does not scale well with the number of classes. The paper may perform an experiment to ablate this design decision.
* It seems the proposed technique requires two forward passes to compute $\theta$ and $\theta_{desc}$. The same applies to the text embeddings. This can be made clearer in Section 4.3
* The Out-of-Vocabulary benchmark replaces class names with class descriptors and is tested on the same 11 datasets as the other benchmarks. Then the term Out-of-Vocabulary may be misleading, since the classes are actually “in-vocabulary”. It may make sense to just name this benchmark as a “Class Descriptor classification” or something similar.

### Minor comments:
* What does naively combine “global” and “local features” mean?
* The learnable bias adds a fixed vector across all positions. So IMO this is global rather than local.
* The paper may add the clip baseline to fig 5. It can be interesting to see if the clip model has this localization property but loses it during finetuning.

**Strengths And Weaknesses:**

### Strengths:

* The proposed technique achieves strong performance on a number of benchmarks as compared to the baselines.
* The paper combines a number of different techniques in finetuning VLM’s and achieves strong performance.

### Weaknesses:

* The major weakness of the paper is the role of "semantical alignment”. IMO, the ablations do not convincingly back up the importance of this component.
* More details of the proposed model can be added.

---

> ### Author Response · Authors · 2024-09-07
> **Response to Reviewer xQ13**
>
> Thank you for your thoughtful comments and feedback. We have responded to each of your comments/suggestions below. We have also summarized our core contributions in this work in the common response to all reviewers.
>
> > 1) The major weakness of the paper is the role of "semantical alignment”. IMO, the ablations do not convincingly back up the importance of this component.
>
> There may be a misunderstanding here -- we use the term "*semantic alignment*" to indicate how text semantics via class descriptions can be integrated into both text and visual modality inputs of a VLM, and how this provides improved generalization performance across multiple benchmark tasks.
>
> * Our definition of semantic alignment is provided in Eqn 1 of the paper, which as stated takes an **average of similarities between decription-guided text features and image features**. We do not claim that local image features alone are the primary reason for our improved generalization. We have also clarified this in the revised manuscript (Sec 1, Paragraph 3 and Sec 4, Paragraph 1).
>
> * As discussed in Section 5.3, our definition of semantic alignment is critical for the performance improvement. Our methodology has provisions for incorporating semantics in both modalities. While the contribution of _description-guided text features_ is higher, the parameter-free cross-attention module provides a computationally inexpensive mechanism to integrate semantics into the image modality. This provides an improvement of ~0.4% in our results at a low cost. We note that it is common for methods in this space to show a max of 1% improvement over the previous state-of-the-art; for example, PromptSRC showed a gain of 0.96% over MaPLe for novel classes. In this light, a 0.4% improvement obtained using the image modality is also non-trivial.
>
> * We add the bias term to enhance the learning of local features, and we have emphasized its contributions more clearly in the revised manuscript (Sec 4, Paragraph 1 and in Section D of the Appendix). To elaborate on the necessity and role of the bias:
>
> 1) Since CLIP is originally trained using only the CLS token, our initial attempts to use the patch tokens directly to compute local features resulted in poor performance due to pretrained layer norms (see Figure 3 in the paper). Therefore, we opted to adjust the local features in a parameter-efficient manner by adding a learnable bias. This helps in obtaining better local features.
> 2) However, the bias alone is not sufficient; it must operate alongside cross-attention. The bias refines local features, while cross-attention integrates descriptive information into the visual domain, using local features.
>
>
> > 2) Further details of how hyperparameters , and the number of soft prompts are tuned may be reported. This could be contrasted with the different baseline methods
>
> In the implementation details section of the paper (Appendix Sec C), we provide a comprehensive description of the hyperparameters used. We use the same configuration for all datasets. For fair comparison with baseline methods, we avoid detailed hyperparameter tuning techniques or mixed precision training; the improvements presented in our results come directly as a consequence of our methodological contributions in using only class descriptions and object-level features to perform semantic alignment and not from any other hyperparameter choices beyond the baselines. (We follow the hyperparameters of the strongest existing baseline PromptSRC.)
>
> As reported in Appendix Section C, we tune 4 prompt vectors for each transformer layer, for 9 layers in each encoder. Table 1 shows that we tune fewer parameters than many baselines.
>
> > 3. How is $\alpha$ chosen?
>
> We describe a simple heuristic for alpha. As explained in Section 4.3, for each description, the maximum attention weight over image patches is a proxy for the specificity of the description. We then define $\alpha$ as the average specificity for all descriptions. We do not treat it as a hyperparameter.

---

> ### Author Response · Authors · 2024-09-07
> **[Contd] Response to Reviewer xQ13**
>
> > 4. While finetuning the VLM, the class-descriptions of the ground-truth class are readily available. So, to compute , one can just use the class-decriptions of the ground-truth class instead of using all class descriptions . Using all class descriptions is quite noisy and also does not scale well with the number of classes. The paper may perform an experiment to ablate this design decision.
>
> Using class descriptions of the ground-truth class makes sense during training but may lead to noisy local features at inference. Our intention of using class descriptions of all _training classes_, is to construct a generalizable local view of the image, rather than a biased one. Due to the unbiased nature of the feature, it can help with tasks like Out-of-Vocabulary Classification.  The table below shows the impact of using just the ground-truth class descriptions during training on three benchmarks. We do not change any hyperparameters. These results corroborate our perspective.
>
> **Out-of-Vocabulary Classification Benchmark**:
> || Avg Base Acc| Avg Novel Acc|Avg HM |
> | --| ---|---|---|
> | all descriptions (Ours) | 43.30 | 45.60| 44.40|
> | ground truth descriptions | 41.76 | 43.45 | 42.59|
>
> **Base-to-Novel Generalization Benchmark**:
> || Avg Base Acc| Avg Novel Acc|Avg HM |
> | --| ---|---|---|
> | all descriptions (Ours) | 84.68 | 77.51| 80.94|
> | ground truth descriptions | 84.58 | 76.93 | 80.58|
>
> **Generalized Zero-Shot Benchmark**:
> || Avg Base Acc| Avg Novel Acc|Avg HM |
> | --| ---|---|---|
> | all descriptions (Ours) | 79.46 | 69.75| 74.29|
> | ground truth descriptions | 79.27 | 68.96 | 73.76|
>
>
>
> > 5) It seems the proposed technique requires two forward passes to compute $\theta$ and $\theta_{desc}$. The same applies to the text embeddings. This can be made clearer in Section 4.3
>
> We only require a single forward pass to compute $\theta$ and $\theta_{desc}$. As described in Sec 4.3 and Figure 3, only the $cls$ token is used for image features. We use the remaining tokens to compute local image features $\theta_l$, and consequently $\theta_{desc}$ (instead of discarding the tokens, as is typically done). We have clarified this in Section 4.3 of the revised manuscript. For the steering losses, we precompute the zero-shot embeddings and cache them.
>
> > 6) The Out-of-Vocabulary benchmark replaces class names with class descriptors and is tested on the same 11 datasets as the other benchmarks. Then the term Out-of-Vocabulary may be misleading, since the classes are actually “in- vocabulary”. It may make sense to just name this benchmark as a “Class Descriptor classification” or something similar.
>
> Thank you for the suggestion. We designed our experiment using standard datasets owing to the paucity of datasets which are actually out-of-vocabulary for CLIP. For better clarity we have modified this to "Classification without Class-names (CwC)" in the revised manuscript. We hope this is clearer.
>
> ### minor comments
>
> > 7. What does naively combine “global” and “local features” mean?
>
> We generate $\theta_{desc}$ by incorporating class descriptions into local image features. In Sec 5.3, to study the goodness of incorporating class descriptions, we compare against a baseline which takes an average of global and local image features, without the cross-attention module. We have made this clear in the revised manuscript (Sec 5.3).
>
>
> > 8. The learnable bias adds a fixed vector across all positions. So IMO this is global rather than local.
>
> The bias vector is indeed global in context. We use the word local to indicate that the bias is added to facilitate the learning of better local image features.
>
>
> > 9. The paper may add the clip baseline to fig 5. It can be interesting to see if the clip model has this localization property but loses it during finetuning.
>
> Thank you for the suggestion. We have updated Figure 5 with zero-shot CLIP.
>
> Your suggestions have greatly improved the presentation of our paper. We appreciate this opportunity to respond to your queries, and will be happy to answer further as required.

---

> > ### Comment · Reviewer_xQ13 · 2024-09-23
> > **Rebuttal Response**
> >
> > Thanks to the authors for their rebuttal. The experiments comparing class descriptions of the ground-truth class vs all classes are interesting and counter-intuitive. There indeed may be a regularization effect by using all class descriptions. They can consider adding these to the supplementary and reference it in the main article.
> >
> > However,
> >
> > In the current version of the paper, in Table 8, removing the learnable bias reduces the performance by 1% while removing cross-attention between text-features and local image features just reduces it by 0.3%. The authors seem to suggest that the learnable bias is needed along with the proposed semantic alignment module. But the results do not demonstrate that. The authors can report the results of the following baseline: Class Descriptions + learnable bias and without the cross-attention module.
> >
> > Minor:
> >
> > > We only require a single forward pass to compute $\theta$ and $\theta_{desc}$
> >
> > The reason I mention this in my original review was that Algorithm 1 in the supplementary seemed to indicate one requires two forward passes to compute $\theta$ and $\theta_{desc}$. But now I understand, that $\theta$ can be precomputed.

---

> > > ### Author Response · Authors · 2024-09-26
> > > **Further Response to Reviewer xQ13**
> > >
> > > Thank you for responding to our rebuttal, as well as acknowledging the interesting results of our method. We look forward to sharing this with the broader community too.
> > >
> > > 1. We have added the results of the suggested ground truth attribute study to Sec D of the Appendix under "Using Class descriptions of Only Ground Truth Classes". Thank you for this suggestion.
> > >
> > > 2. We would like to clarify that the requested baseline *"class descriptions + learnable bias without cross attention"*, was already evaluated and present in Table 8 of the main paper under the name *SAP w/ global & local* (80.55 HM). We recognize that the absence of the phrase *cross-attention module* may have cause some confusion. As explained in Sec 5.3 under *Effect of Removing Class Descriptions*, the *SAP w/ global & local* ablation naively combines both global and local features (the local features are finetuned using the bias) by averaging them, but does not integrate class descriptions in the image modality through cross-attention. In this setup, class decsriptions are only used in the text modality.
> > >
> > >    To further clarify the effect of the learnable bias, we present a summary of existing results from Table 8 of our paper:
> > >
> > >    |Class Descriptions | Bias for local features | Cross   Attention|Avg. Novel | Avg HM|
> > >    | -| -|-|-| -|
> > >    |✓|✓|✓|77.51|80.94|
> > >    |✓|✓|[x]|76.81|80.55|
> > >    |✓|[x]|✓|75.72|79.9|
> > >    |✓|[x]|[x]|77.04|80.63|
> > >
> > >    The Class Descriptions column in the table indicates their usage in the text modality. The table highlights that directly augmenting global image features with bias fine-tuned local features without cross-attention (row 2) performs worse than using just the global image features (row 4). This indicates that the bias alone is insufficient, and best results are obtained when the bias fine-tuned local features are integrated with class descriptions using our cross-attention module (row 1).
> > >
> > >    Additionally, the result in row 3 suggest that without a learnable bias, the local features become uninformative, even when cross-attention is used. This indicates that the learnable bias plays a critical role in making the local features useful for semantic alignment. We have added these clarifications to Sec D of the Appendix under "Role of the Learnable Bias Term" of the revised manuscript.
> > >
> > > 3) To further support the importance of the cross-attention and bias together, we studied the performance drop when both the cross-attention and bias are removed across different architectures and settings. The results are present in the table below:
> > >
> > >    |Architecture-Setting | Avg. Novel | Avg. HM|
> > >    |-|-|-|
> > >    |ViT-B2N|-0.47|-0.31|
> > >    |RN50-B2N|-0.49|-0.35|
> > >    |ViT-GZS|-0.49| -0.53|
> > >    |RN50-GZS|-0.40|-0.31|
> > >
> > >    The bias and cross-attention modules are complementary, with both aiding in incorporating class descriptions into the visual modality. We have also added this table to the revision in Sec D of the Appendix under "Effect of Removing Class Descriptions from the Image Modality".
> > >
> > >
> > > We hope that the tables above clarify the role of the bias term and its complementary interaction with the cross-attention module.
> > >
> > > We'd be happy to discuss further and provide any further clarifications.

---

### Review · Reviewer_uP8W · 2024-08-18

**Summary Of Contributions:**

Vision-Language Models (VLMs) perform image classification on a downstream dataset by comparing an image representation with text representations of the class names in the dataset’s label space. It has been shown that, when a small amount of labeled data is available,
 fine-tuning VLMs substantially boosts downstream performance. However, the fine-tuned model does not generalize to novel classes that were absent during fine-tuning. This work proposes Semantic Alignment for Prompt Learning (SAP). The key technique of SAP is that it leverages class descriptions to fine-tune VLMs for better generalization to novel classes. Empirical evaluations across various benchmarks and tasks demonstrate that SAP can effectively improve VLMs' performance.

**Audience:**

Yes

**Broader Impact Concerns:**

There is no concern on the ethical implications of the work.

**Claims And Evidence:**

Yes

**Requested Changes:**

Based on the "Weaknesses" section, it would be great if the authors could include:
- SAP's performance under various LLMs generated label descriptions.
- The costs (computation or API querying prices) for generating the label descriptions.

**Strengths And Weaknesses:**

Strengths:
- The paper is generally well-written and well-motivated.
- Improving the fine-tuning methods of VLMs is of practical importance.
- The proposed SAP approach is conceptually sound.
- The empirical results of the proposed SAP are very promising.

Weaknesses:
- The proposed SAP approach leverages an LLM to generate label descriptions, which seems to heavily rely on the LLM's capabilities. It is unclear how powerful an LLM is required. This appears to necessitate the LLM having very strong knowledge of basic physical world concepts. Additionally, the extra costs for label description generation are not clearly discussed.
- Using auxiliary knowledge to enhance VLMs (or foundation models in general) seems to be a common practice, which diminishes the novelty of the proposed method.

---

> ### Author Response · Authors · 2024-09-07
> **Response to Reviewer uP8W**
>
> Thank you for your valuable comments and feedback. We have responded to each of your comments/suggestions below. We have also summarized our core contributions in this work in the common response to all reviewers.
>
> > 1. The proposed SAP approach leverages an LLM to generate label descriptions, which seems to heavily rely on the LLM's capabilities. It is unclear how powerful an LLM is required. This appears to necessitate the LLM having very strong knowledge of basic physical world concepts. Additionally, the extra costs for label description generation are not clearly discussed.
>
> There has been recent research indicating that LLMs can be used as a cheap source of generic world knowledge [1]. We use GPT-3.5, which is a relatively weak LLM (when compared to more recent LLMs) and is relatively inexpensive to query. Many similarly powerful open-source models also exist [2], which can be used to generate relevant class descriptions.
>
> As explained in Section 4.1, we *generate class descriptions at a class name level, not for individual images*. This makes our approach scalable -- even for every test image, we use the same class descriptions. We also do not perform any postprocessing on the generated descriptions. **Generating the class descriptions for all 11 datasets and 4000 classes considered across our experiments takes ~1 hour, and costs less than $1.** We also believe that access to relatively stronger LLMs may only improve in the near future, making this approach more tenable. We have added this discussion in the revised manuscript Section D of the Appendix.
>
> To study this further, we considered other LLMs (beyond what we had in our earlier version), and find that the impact of the specific LLM used is negligible.
> In particular, we generated class descriptions from two other LLMs - OpenAI's GPT4o-mini and Anthropic's Claude Haiku. Both LLMs considered are fast and cheap -- for instance, *generating class descriptions from Claude Haiku takes ~40 mins and costs $0.5 for all 11 datasets and 4000 class labels considered across our experiments*. Our results are presented in the table below:
>
> **Generalized Zero-Shot Benchmark**
> | LLM | Avg Base Acc | Avg Novel Acc | Avg HM |
> | ---- | ---------- | --------------| --------|
> |GPT-3.5 | 79.47 |  69.75| 74.29 |
> | Claude Haiku | 79.31 | 69.14 | 73.88 |
> | GPT4o-mini |79.54 |69.53  | 74.2|
>
> **Base-to-Novel Generalization Benchmark**
> | LLM | Avg Base Acc | Avg Novel Acc | Avg HM |
> | ---- | ---------- | --------------| --------|
> |GPT-3.5 | 84.68 | 77.51 | 80.94 |
> | Claude Haiku | 84.64 | 77.05 | 80.67 |
> | GPT4o-mini |84.74 | 77.16 | 80.77|
>
> The results indicate that we get similar results across varying quality of outputs from different LLMs; we however acknowledge that the LLM must be one that is capable of generating informative descriptions. We add these results to Section D of the Appendix in the revision.
>
> We also study how uninformative descriptions may impact the performance of our method. We replace class decriptions with random words, and we find that this leads to a significant drop in performance as shown in Table 10 of Appendix Section D. We hence infer that a reasonable level of text semantics obtained from class labels can be very useful for learning prompt tuning methods in adaptation of VLMs for classification.
>
> > 2. Using auxiliary knowledge to enhance VLMs (or foundation models in general) seems to be a common practice, which diminishes the novelty of the proposed method.
>
> As mentioned in the common response to the reviewers, our contribution is situated among methods that focus on *learnable prompt tuning for adaptation of VLMs* (e.g. CoOp, CoCoOp, PromptSRC, MaPLe, KgCoOp, CoPrompt). Our method is the first in this line of work to bring image regions (at a part level) with concept-level semantics derived from class labels. Our methodological contribution comes from how such descriptions can be integrated with both text and visual modalities using the notions of semantic alignment and cross-attention.
>
> This approach achieves superior transferability to new classes and datasets -- especially on the more challenging setting of generalized zero-shot classification that we introduce in this work. Our approach also shows enhanced performance in larger label spaces. In Sec 5.3 of the paper, we study the impact of our design choices, namely i) how to incorporate semantics into both modalities and ii) how to compute semantic alignment. Table 7 shows that our design choices play an important role in the strong performance.

---

> > ### Author Response · Authors · 2024-09-07
> > **[Contd] Response to Reviewer uP8W**
> >
> > ### Requested Changes:
> >
> > > 3. SAP's performance under various LLMs generated label descriptions.
> >
> > Please see the first response above. As shown in the table therein, we show the performance of SAP on Claude Haiku and GPT4o-mini. We have added this discussion in the revised manuscript Section D of the Appendix.
> >
> > > 4. The costs (computation or API querying prices) for generating the label descriptions.
> >
> > We query GPT-3.5 which is relatively inexpensive. Since we require descriptions only at the class level, the time taken to generate all descriptions for all datasets is close to 1 hour, and costs less than $1.
> >
> > Your suggestions have greatly improved the presentation of our paper. We appreciate this opportunity to respond to your queries, and will be happy to answer further as required.
> >
> > References:
> > [1] Korinek, A. LLMs Level Up—Better, Faster, Cheaper: June 2024 Update to Section 3 of “Generative AI for Economic Research: Use Cases and Implications for Economists,” Published in the. Journal of Economic Literature, 61, 4.
> >
> > [2] Zhao, W. X., Zhou, K., Li, J., Tang, T., Wang, X., Hou, Y., ... & Wen, J. R. (2023). A survey of large language models. arXiv preprint arXiv:2303.18223.

---

### Author Response · Authors · 2024-09-07
**Common Response to All Reviewers**

We thank all reviewers for their thoughtful feedback. We are pleased to see the following encouraging comments from the reviewers:

1. The proposed method is sound and has practical importance (uP8W).
2. The proposed method of fine-tuning VLMs leads to strong empirical results (uP8W, xQ13, 3NEX).
3. The two newly proposed evaluation schemes are useful (3NEX).
4. The paper is well-written and well-motivated (uP8W).

As a summary, our key contributions lie in expanding upon the recent efforts on learnable prompt tuning methods for adapting foundational vision-language models (e.g. CoOp, CoCoOp, PromptSRC, MaPLe, KgCoOp, CoPrompt) to a diverse set of tasks. Our method is the first in this line of work (learnable prompt tuning methods for adapting VLMs) to bring image regions (at a part level) with concept-level semantics derived from class labels. This approach achieves superior transferability to new classes and datasets -- especially on the more challenging setting of generalized zero-shot classification that we introduce in this work. Our approach also shows enhanced performance in larger label spaces. We believe that a venue like TMLR that encourages new ideas is an ideal venue for this work.

We address each reviewer’s concerns individually below. We have also uploaded the revised manuscript with the suggested changes. The changes are highlighted in red.

---

### Author Response · Authors · 2024-09-20
**Gentle reminder to discuss**

Dear Reviewers,

Since the two-week window for discussions is coming to a close, we kindly request you acknowledge our rebuttal and provide us an opportunity to address any remaining concerns you may have. Thank you.

---

### Decision · Action_Editor_eeTL · 2024-11-04

**Recommendation:** Accept with minor revision

**Comment:**

Most reviewers are positive about the paper.

One Reviewer noted that Table 8 showed only a minimal impact of cross-attention (+0.3%), whereas the learnable bias had a larger effect (+1.0%) which weakens a bit the claim of the paper of the importance of semantic alignment. I suggest the authors to elaborate more on this point in the final version: I think one way to improve the paper is to pinpoint really what is the most important component in the new architecture. I think the authors might move the clarifications added in Sec D of the Appendix under "Role of the Learnable Bias Term" earlier in the main paper. Another source of confusion was the fact that the lowest performance removing both bias and cross-attention module in the Table provided in the rebuttal is 79.9 (thus a drop of 1.0%), while the overall approach outperforms the baselines by 1.4%. There have been some questions as to why the discrepancy (what justifies the +0.4% wrt the baselines?) which I hope the authors will elaborate on in the paper. Overall, provided the authors clarify these aspects, the paper extensively demonstrates the efficacy of the overall architecture.

**Audience:**

multi-modal few-shot learning audiences might find this paper interesting

**Claims And Evidence:**

This paper introduces a prompt-tuning approach for VLMs. The method generates class descriptions from LLMs and include them in a clip-style contrastive training architecture, which reduces overfitting in low-shot training scenarios and improves generalization to unseen classes. The authors evaluate the method across 11 benchmark datasets, introducing two new evaluation protocols: Generalized Zero-Shot Classification and Classification without Class-names (CwC).

The proposed architecture appears to outperform a number of baselines in the benchmark tested (+1.4 / +1.0). All reviewers are convinced by the observed improvement. However, some of the claims around the importance of the semantic component generated confusion for some of the reviewers. The paper claims SAP’s robustness to variations in LLM quality for generating class descriptions. This is well-supported by results using different LLMs added after the rebuttal, showing similar performance across models of varying strength.

---

> ### Author Response · Authors · 2024-12-02
> **Camera-Ready Submission**
>
> We once again thank all reviewers and the AE for the constructive feedback, which has helped improve the presentation of our paper. As suggested by the AE, we have moved the study on the learnable bias from the Appendix to Section 5.2 of the main paper under the heading "Role of the Learnable Bias Term".
>
> Besides this, we state here the changes made in earlier versions, for completeness:
>
> - We have added various clarifications in response to reviewer comments in Section 4 of the main paper
> - In Section D of the Appendix, we have added results for two other LLMs -- OpenAI's GPT4o-mini and Anthropic's Claude Haiku
> - Results of using just the ground-truth class-descriptions have also been added to Section D of the Appendix